# Fear extinction is regulated by the activity of long noncoding RNAs at the synapse

Wei-Siang Liau [1,12] ✉, Qiongyi Zhao [1,12], Adekunle Bademosi [2], Rachel S. Gormal [2], Hao Gong [1], Paul R. Marshall[1], Ambika Periyakaruppiah[1], Sachithrani U. Madugalle[1], Esmi L. Zajaczkowski [1], Laura J. Leighton [1], Haobin Ren [1], Mason Musgrove [1], Joshua Davies[1], Simone Rauch[3], Chuan He [3], Bryan C. Dickinson [3], Xiang Li[4,5], Wei Wei[4], Frédéric A. Meunier [2,6], Sandra M. Fernández-Moya [7,11], Michael A. Kiebler [7], Balakumar Srinivasan[8], Sourav Banerjee [8], Michael Clark [9], Robert C. Spitale [10] & Timothy W. Bredy [1] ✉

Long noncoding RNAs (lncRNAs) represent a multidimensional class of regulatory molecules that are involved in many aspects of brain function. Emerging evidence indicates that lncRNAs are localized to the synapse; however, a direct role for their activity in this subcellular compartment in memory formation has yet to be demonstrated. Using lncRNA capture-seq, we identified a specific set of lncRNAs that accumulate in the synaptic compartment within the infralimbic prefrontal cortex of adult male C57/Bl6 mice. Among these was a splice variant related to the stress-associated lncRNA, *Gas5*. RNA immunoprecipitation followed by mass spectrometry and single-molecule imaging revealed that this *Gas5* isoform, in association with the RNA binding proteins G3BP2 and CAPRIN1, regulates the activity-dependent trafficking and clustering of RNA granules. In addition, we found that cell-type-specific, activity-dependent, and synapse-specific knockdown of the *Gas5* variant led to impaired fear extinction memory. These findings identify a new mechanism of fear extinction that involves the dynamic interaction between local lncRNA activity and RNA condensates in the synaptic compartment.

The extinction of conditioned fear is a critically important adaptive behavior that is driven by synaptic activity in the infralimbic prefrontal cortex (ILPFC). Like other forms of learning, long-lasting memory for fear extinction depends on coordinated changes in gene expression[1–3].

Although significant progress has been made in revealing the mechanisms that regulate this process, a complete understanding of the molecular code underlying the formation of fear extinction memory is still lacking. Long noncoding RNAs (lncRNAs) have emerged

[1]Cognitive Neuroepigenetics Laboratory, Queensland Brain Institute, The University of Queensland, Brisbane, QLD, Australia. [2]Single Molecule Neuroscience Laboratory, Queensland Brain Institute, The University of Queensland, Brisbane, QLD, Australia. [3]Department of Chemistry, The University of Chicago, Chicago, IL, USA. [4]Department of Neurosurgery, Zhongnan Hospital of Wuhan University, Wuhan, China. [5]Medical Research Institute, Wuhan University, Wuhan, China. [6]School of Biomedical Sciences, The University of Queensland, Brisbane, QLD, Australia. [7]Biomedical Centre, Ludwig Maximilian University of Munich, Munich, Germany. [8]National Brain Research Centre, Manesar, India. [9]Department of Anatomy and Physiology, University of Melbourne, Parkville, VIC, Australia. [10]Department of Pharmaceutical Sciences, The University of California, Irvine, CA, USA. [11]Present address: Gene Regulation of Cell Identity, Regenerative Medicine Program, Bellvitge Institute for Biomedical Research (IDIBELL) and Program for Advancing Clinical Translation of Regenerative Medicine of Catalonia, P-CMR[C], L'Hospitalet de Llobregat, 08908 Barcelona, Spain. [12]These authors contributed equally: Wei-Siang Liau, Qiongyi Zhao. ✉e-mail: w.liau@uq.edu.au; t.bredy@uq.edu.au

as key regulatory molecules associated with a variety of important biological processes, including gene regulation, translation, and RNA trafficking[4,5]. Central to their multidimensional functions, lncRNAs are commonly expressed in a cell-type-specific and spatiotemporal manner and are highly enriched in the brain[6]. Several lncRNAs have been found to accumulate in the synaptic compartment in response to neural activity. For example, the lncRNA ADEPTR accumulates in dendrites where it mediates activity-dependent changes in synaptic plasticity[7,8], suggesting a potential role for the localized expression of lncRNAs in the regulation of synaptic processes underlying memory.

We previously found activity-dependent epigenetic regulation of gene expression that is associated with fear-related learning and modulated, in part, by nuclear lncRNAs[9,10]. We therefore queried whether lncRNAs at the synapse are also a key feature of the localized regulation of cellular processes underlying fear extinction. To address this, we used lncRNA capture-sequencing to map the expression of synapse-enriched lncRNAs in the ILPFC of adult male C57/bl6 mice, followed by single-molecule tracking in live cortical neurons and a CRISPR-inspired cell-type- and synapse-specific, and state-dependent RNA knockdown approach. This revealed the critical involvement of a variant of the lncRNA *Gas5* in the trafficking of RNA granules at the synapse, intrinsic neuronal excitability, and the formation of fear extinction memory.

## Results

### A significant number of lncRNAs are enriched at the synapse in the adult ILPFC

It has previously been shown that lncRNAs are abundantly expressed in different subcompartments of the cell, including the nucleus, nucleolus, and paraspeckles where they coordinate key cellular processes such as chromatin state and splicing, and can serve as decoys for other regulatory RNAs in the cytoplasmic compartment[11–17]. However, very few studies have examined lncRNA activity in the synaptic compartment[7,18,19] and none has examined their compartment-specific role in the context of learning and memory. To begin to determine whether there are synapse-enriched lncRNAs that are involved in fear extinction learning, we employed synaptosome isolation followed by lncRNA capture sequencing on tissue derived from the adult mouse ILPFC of a pool of samples derived from both retention control (RC) and extinction (EXT) trained mice (Fig. 1a). To verify the purity of the synaptosome preparation, we used markers for the dendritically localized scaffold protein PSD-95, synaptophysin, and the nucleus-specific chromatin modifier, HDAC2 (Supplementary Fig. 1a–c). For targeted lncRNA enrichment using capture-seq, we employed a panel of 190,689 probes that tiled across 28,228 known and predicted lncRNA transcript isoforms[20] (Supplementary Data 4 and Supplementary Data 5). Based on this approach, we identified 30,187 expressed lncRNA transcripts, with similar numbers of genes and transcripts expressed in the nucleus[10] and synapse of ILPFC neurons (Fig. 1b and Supplementary Data 1). Of these, 2583 were enriched in the synaptic compartment, including many predicted (62.5%, 1615) and annotated (GENCODE V25, 37.5%, 968) lncRNAs (Supplementary Data 2), with the largest proportions being derived from intragenic (51.2%, 1324), extragenic (28.7%, 741) and antisense (18.7%, 483) regions (Fig. 1c, Supplementary Data 2). Among the 2583 synapse-enriched lncRNAs, 88 lncRNAs were unique to the synaptic compartment (i.e. 0 reads in nucleus samples). However, all 88 lncRNAs exhibited a very low level of expression, with the majority showing within-group variability (i.e. 1-2 replicates show no expression while the other replicates show low expression). Therefore, we sorted the synapse-enriched lncRNAs by their average FPKM and focused on synapse-enriched lncRNAs that exhibited a relatively high FPKM in the synaptic compartment (Fig. 1d).

In contrast to our recent work on enhancer-derived lncRNAs (eRNAs) and memory, which revealed 434 eRNAs directly associated with fear extinction[10], we identified 35 putative eRNAs at the synapse, suggesting that the majority of synapse-enriched lncRNAs do not belong to this subclass of noncoding RNA. Furthermore, while the majority of synapse-enriched lncRNAs (76.9%, 1987) contained putative transposable elements, including both short interspersed nuclear elements (SINEs) and long interspersed nuclear elements (LINEs) (Supplementary Data 2) Synapse-enriched lncRNAs did not exhibit enrichment of SINE or LINE elements compared to nucleus-enriched lncRNAs (Supplementary Data 2).

To determine if the levels of these highly-abundant synapse-enriched lncRNAs were altered by fear extinction training (EXT), we next selected 8 of the top 10 synapse-enriched lncRNAs for testing by RT-qPCR. In an independent cohort of behaviorally trained mice, a comparison between mice that had been fear conditioned followed by exposure to a novel context 24 h later with no further cue exposure (retention control, RC) and mice that had been fear conditioned followed by fear extinction training (EXT) revealed a fear extinction learning-induced increase in the expression of 7 lncRNAs: *Rpph1*, *Rn7sk*, *Rmrp*, 93301121K16Rik, Gm28437, Gm47305 and *Gas5* (Fig. 1e–l). This increase was specific for fear extinction learning as a single retrieval cue did not alter their expression at the synapse (Supplementary Fig. 2b). At the whole transcriptome level using a standard RNA sequencing approach on synaptosome-derived RNA from RC and EXT trained mice, we were unable to detect a statistically significant change in learning-induced candidate lncRNA expression, although 6 out of 8 exhibited a trend (Supplementary Data 3). These data highlight the advantage of using a more sensitive targeted sequencing approach to detect lncRNA activity within a heterogeneous cell population. Nonetheless, among the lncRNAs identified by capture-seq and subsequently shown to be upregulated by fear extinction training (EXT) by RT-qPCR, the stress-responsive lncRNA *Gas5* attracted our attention as it has been implicated in the regulation of motivated behavior[21,22]. Given that fear extinction is associated with changes in stress reactivity and involves prefrontal cortex-dependent learning, we focused our subsequent investigation on the functional relevance of *Gas5* in fear extinction memory.

### An alternative splice variant of *Gas5* is enriched at the synapse and is associated with fear extinction

An initial analysis of alternative splicing (Fig. 2a) associated with lncRNA localization revealed a significantly higher proportion of skipped exons in synapse-enriched lncRNAs compared to nucleus-enriched lncRNAs (32% in the synapse, 17% in the nucleus, $p = 2.41e^{-45}$) (Fig. 2b top). In contrast, nucleus-enriched lncRNAs exhibited greater intron retention (21% in the nucleus, 8% in the synapse, $p = 2.13e^{-52}$) (Fig. 2b bottom), in agreement with there being partially spliced RNA intermediates in the nucleus, as well intron-retaining functional nuclear RNAs such as an isoform of *Tug1*[23]. Upon closer analysis of the sequencing data, we found differential expression of individual *Gas5* splice variants in the synapse compared to the nucleus (Fig. 2d, Supplementary Fig. 3). In contrast to intron-retained *Gas5* variants (Supplementary Fig. 3), which were confined to the nucleus, the *Gas5* variant ENSMUST00000162558.7 was the most highly enriched isoform in the synaptic compartment (Fig. 2e, f, Supplementary Data 3). As a control, we also examined the localized expression of *Meg3* as it is activity-dependent, associated with synaptic plasticity, and has previously been shown to be selectively enriched in the nucleus[24]. As expected, the *Meg3* transcript was preferentially expressed in the nuclear compartment (Fig. 2f). The expression of the *Gas5* variant in dendrites was confirmed using RNAscope in primary cortical neurons, in vitro (Fig. 2g). In addition, we also observed that the *Gas5* variant co-localizes with PSD95 (ratio of co-localized dendritic puncta: −KCl, 5.76; +KCl, 3.02) and the dendritic marker, SV2A (ratio of co-localized dendritic puncta: −KCl, 2.46, +KCl, 2.98) in dendrites (Supplementary Fig. 4). Taken together, the findings suggest that, in the adult brain, a specific *Gas5* variant can localize to the synapse in an experience-dependent manner.

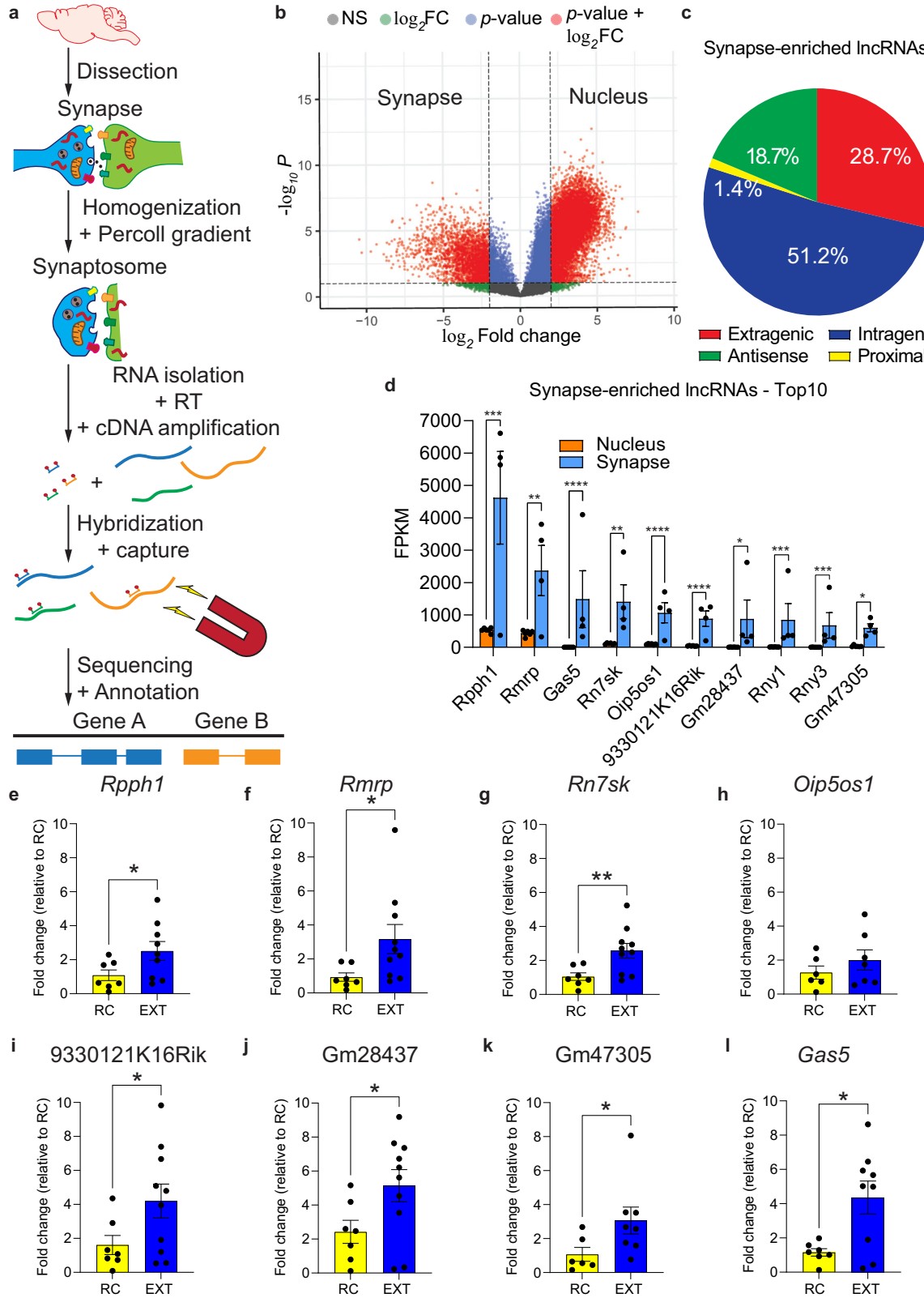

## The synapse-enriched *Gas5* variant interacts with RNA binding proteins involved in translation and RNA localization, as well as RNA granules

To begin to explore the *Gas5*-protein interaction network in the ILPFC and its relationship with behavioral experience, we performed RNA immunoprecipitation followed by mass spectrometry using a synthetically designed biotinylated *Gas5* variant to pull down synapse-enriched lncRNA:protein complexes in ILPFC samples derived from RC and EXT mice. Overall, the *Gas5* variant was observed to interact with 524 proteins at the synapse, with the majority (85%, 418) found in both the RC and EXT groups. Within the RC group, we detected 67 (12%) unique proteins, whereas 39 (8%) proteins were specific to the EXT group (Fig. 3a, Supplementary Data 6).

**Fig. 1 | Targeted RNA capture-seq reveals a myriad of synapse-enriched lncRNAs. a** Schematic overview of the lncRNA capture-seq. **b** Volcano plot showing all nuclear and synaptic lncRNAs expressed (FPKM > 0.5) in the ILPFC (Ballgown parametric F-test). Significantly enriched transcripts are those with a fold change of ≥4 and a *p*-value of ≤0.05. Significant hits are highlighted in red. **c** Classification of captured synaptic lncRNAs based on their genomic location with respect to protein coding genes and according to GENCODE V25 annotation. **d** Bar plots showing the top 10 lncRNAs that are significantly enriched at the synapse and expressed as Fragments Per Kilobase of transcript per Million mapped reads (FPKM) (Supplementary Data 2) (nucleus and synapse, *n* = 6 and 4 independent biological replicates per group, respectively, Ballgown parametric F-test). *Rpph1*, *q* = 0.001544; *Rmrp*, *q* = 0.006962; *Gas5*, *q* = 0.000628; *Rn7sk*, *q* = 0.008848; *OipSos1*, *q* = 0.000161; 9330121K16Rik, *q* = 0.00001; Gm28437, *q* = 0.010893; *Rny1*,

*q* = 0.004098; *Rny3*, *q* = 0.003025; Gm47305, *q* = 0.013784. *\**q* < 0.05, **\*q* < 0.01, ***\*q* < 0.005, ****\*q* < 0.001. Error bars represent S.E.M. **e–l** RT-qPCR of 8 of the 10 synapse-enriched candidates in the ILPFC following fear extinction training. 18S rRNA was used as the housekeeping gene for normalization (Supplemental Fig. 2a). Statistical significance was determined using two-tailed unpaired Student's *t* test on independent biological replicates (*Rpph1*, RC, *n* = 7, EXT, *n* = 9, t(12.24) = 2.227, *p* = 0.0454; *Rmrp*, RC, *n* = 7, EXT, *n* = 10, t(10.41) = 2.5, *p* = 0.0306; *Rn7sk*, RC, *n* = 7, EXT, *n* = 10, t(13) = 3.159, *p* = 0.0075; *OipSos1*, RC, *n* = 6, EXT, *n* = 7, t(11) = 1.001, *p* = 0.3385; 9330121K16Rik, RC, *n* = 7, EXT, *n* = 10, t(13.55) = 2.25, *p* = 0.0416; Gm28437, RC, *n* = 7, EXT, *n* = 10, t(14.75) = 2.328, *p* = 0.0346; Gm47305, RC, *n* = 6, EXT, *n* = 8, t(10.23) = 2.233, *p* = 0.049; *Gas5*, RC, *n* = 7, EXT, *n* = 9, t(8.814) = 3.263, *p* = 0.0101). *\*p* < 0.05, **\*p* < 0.01. Error bars represent S.E.M.

We next selected CAPRIN1 and G3BP2 for further analysis because of their known roles in the assembly of RNA granules, RNA trafficking, and local translation[25,26]. We first confirmed that *Gas5* interacts with CAPRIN1 and G3BP2 protein in primary FLAG-tagged CAPRIN1 and G3BP2 (Supplementary Fig. 5a, b). We observe that the negative control ADRAM and *Neat* do not bind CAPRIN1 nor G3BP2 in primary cortical neurons. In addition, using RNA immunoprecipitation, we confirmed that CAPRIN1 directly bound to *Gas5* lncRNA in ILPFC tissue derived from both RC and EXT trained mice (Supplementary Fig. 5c). To determine the structural module important for CAPRIN1 and G3BP2 binding, we generated a series of *Gas5* oligonucleotides with deletions tiled across a 504-nucleotide sequence spanning the splice sites for exons 1 through 12 and used these to immunoprecipitate CAPRIN1 or G3BP2 (Fig. 4a). These mutations did not result in the formation of super-stable structures (Supplementary Fig. 6a, b) and the mutant RNAs exhibited minimal degradation after incubation with protein lysates (Supplementary Fig. 6c). ADRAM, a nuclear eRNA involved in mediating epigenetic regulation[10], and *Neat*, a nuclear lncRNA involved in paraspeckles formation[27], exhibited no binding affinity for CAPRIN1 or G3BP2 (Supplementary Fig. 7b, c). A decrease in CAPRIN1 binding was observed when the 3′ terminal end of *Gas5* (408–504 base pairs) was deleted, suggesting that this region of the *Gas5* variant contains the module that is critical for the *Gas5*-CAPRIN1 interaction (Fig. 4b, c, Supplementary Fig. 7). In contrast, a significant reduction in G3BP2 binding occurred when the regions between nucleotides 204–256 and 458–504 were deleted (Fig. 4b, d, Supplementary Fig. 7). Because other *Gas5* splice variants share a similar sequence in the 3′ terminal end of ENSMUST 00000162558.7, it is possible that CAPRIN1 and G3BP2 may also bind to other *Gas5* variants. Furthermore, whether the *Gas5*-CAPRIN1 or *Gas5*-G3BP2 interaction is regulated by RNA modification or dynamic changes in RNA structure remains to be determined. Nonetheless, these findings suggest that specific regions of the *Gas5* variant are critical for its interaction with CAPRIN1 and G3BP2.

### *Gas5* knockdown increases the trafficking of RNA granules and alters their pattern of clustering and assembly

Based on the finding that *Gas5* interacts with RBPs that are involved in RNA trafficking and the formation of RNA granules, including CAPRIN1 and G3BP2, we next investigated the functional relevance of this interaction by examining the effect of *Gas5* knockdown on RNA granule mobility using single-particle tracking photoactivation localization microscopy (sptPALM). We focused on G3BP2 because it forms a core complex with CAPRIN1 and is critically involved in RNA granule trafficking[25]. To visualize and track RNA granules, G3BP2 was fused with the photoconvertible fluorescent protein mEos3.2 and expressed under the control of the human synapsin (*Syn1*) promoter, thereby allowing G3BP2-mEos3.2-specific expression in neurons. We then placed the whole cassette into a lentiviral backbone, packaged the virus and used it to transfect primary cortical neurons. Given that the expression of the *Gas5* variant is increased at the synapse following fear extinction training, we also investigated the functional

consequences of manipulating *Gas5* activity at the synapse, particularly with respect to the trafficking and clustering of G3BP2-containing RNA granules.

To selectively decrease the expression of the *Gas5* variant at the synapse, we used the CRISPR-Cas-inspired RNA targeting system (CIRTS), which is a guide RNA (gRNA)-dependent technology designed to deliver protein cargoes to target RNAs, as its small size facilitates viral packaging and protein delivery to the brain[28]. To target the PIN nuclease effector for RNA degradation at the synapse, we appended the full-length *Calm3* intron as a dendritic localization signal[29] and expressed the CIRTS cassette under the control of a neuron specific *Syn1* promoter on a lentiviral backbone (Fig. 5a). For visualization, GFP was fused upstream of the CIRTS cassette together with a 2A self-cleaving peptide signal. We also determined whether the CIRTS-*Gas5* construct localized to dendrites in a KCl-induced chase experiment in primary cortical neurons. After a 10-minute pulse and chase, an increase in punctate CIRTS-*Gas5* expression was observed in the dendritic spines of transduced neurons, with the accumulation increasing 60 min after the stimulus (Fig. 5b). This indicates that the CIRTS-*Gas5* construct localizes to the synapse and, crucially, does so in an activity-dependent manner. To test the functionality of this system, we designed two gRNAs to specifically target the synapse-enriched *Gas5* variant and examined the effect of each guide in a nuclease cleavage assay (Supplementary Fig. 8a) and in primary cortical neurons (Fig. 5c and Supplementary Fig. 8b). One of the guides degraded the *Gas5* variant ENSMUST00000162558.7 by more than 50% in both the nuclease cleavage assay and in primary cortical neurons, without affecting the expression of other *Gas5* variants and was therefore chosen for all subsequent knockdown experiments (Supplementary Fig. 8).

To test the hypothesis that G3BP2-containing RNA granules are sensitive to neuronal activity and that *Gas5* coordinates their activity in response to stimulation at the synapse, we tracked G3BP2-mEos3.2 in primary cortical neurons transduced with either a CIRTS scrambled control or CIRTS-*Gas5*. Representative images in Fig. 5d show the diffusion of G3BP2 in dendritic regions of interest in control and CIRTS-*Gas5* treated primary cortical neurons. To quantify the nanoscale mobility of G3BP2, we analyzed the mean square displacement (MSD) of the trajectories of individual G3BP2-mEos3.2 molecules. Under low stimulation conditions, *Gas5* knockdown led to a significant increase in mobility, as evidenced by a less constrained and higher area under the MSD curve. This suggests that *Gas5* knockdown altered the trafficking of G3BP2-containing RNA granules (Fig. 5e, f). We next analyzed the clustering of G3BP2-mEos3.2 based on nanoscale analysis using spatiotemporal indexing (NASTIC)[30]. Using this analysis approach, clusters can be identified in single-particle-tracking data as they appear and disappear over time (Fig. 5g). We found that G3BP2-containing RNA granules formed transient clusters with an average radius of approximately 80 nm (Fig. 5h), although there was a modest reduction in CIRTS-*Gas5*-treated neurons. The average cluster lifetime was 5.16 +/− 0.14 s, which increased to 6.18 +/− 0.13 s following *Gas5* knockdown

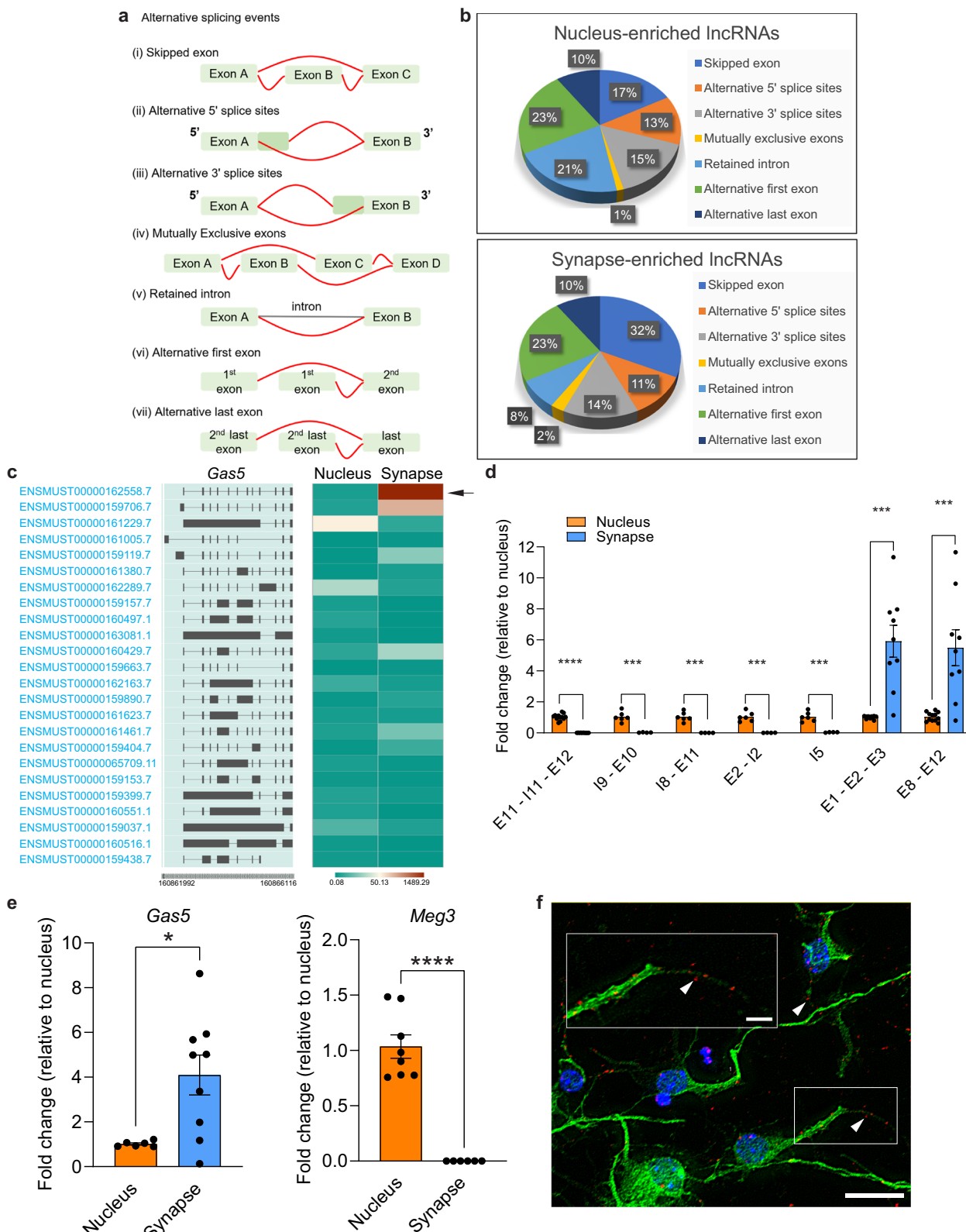

(Fig. 5i), an effect that was accompanied by an increase in the density of G3BP2 within the clusters (Fig. 5j) as well as a reduction in their mobility (Fig. 5k). Given that RNA granule size and molecular assembly are directly linked to transcription, these data suggest that *Gas5* controls both the trafficking and self-assembly/disassembly of G3BP2-containing biomolecular condensates at the synapse. Whether this influences the translational capacity and metabolism of the RNA cargo

held within these ribonucleoprotein complexes, and precisely how this contributes to memory formation, is not yet known.

One possibility is that a change in RNA granule trafficking and clustering influences local synaptic activity. Synaptic plasticity is regulated by the abundance of membrane-bound AMPA receptors (AMPARs) in the postsynaptic compartment[31]. Given that there is an activity-dependent association between *Gas5* and CAPRIN1 within

**Fig. 2 | *Gas5* is highly enriched in the synapse following fear extinction training.**
**a** Illustration of alternative splicing (AS) events. **b** Splicing patterns of AS events in synapse- and nucleus- enriched lncRNAs. Skipped exons (synapse = 32%; nucleus = 17%; two-proportions Z-test, $p = 2.41e^{-45}$), intron retention (synapse = 8%; nucleus = 21%; two-proportions Z-test, $p = 2.13e^{-52}$). **c** Heatmap showing *Gas5* iso-forms expression generated using IsoVis. Red and green indicating high and low expression, respectively. Arrow indicates *Gas5* variant ENSMUST00000162558.7. **d** RT-qPCR of *Gas5* variant expression in the nuclear and synaptic fractions of the ILPFC. The amplified *Gas5* exonic (E) and intronic (I) regions are indicated. *Gas5* transcripts with and without introns are also indicated. Statistical significance was determined using two-tailed unpaired Student's *t* test on independent biological replicates (E11-I11-E12, nucleus, $n = 12$, synapse, $n = 10$, t(11.02) = 16.91, $p < 0.0001$; I9-E10, nucleus, $n = 6$, synapse, $n = 4$, t(5.031) = 7.704, $p = 0.0006$; I8-E11, nucleus, $n = 6$, synapse, $n = 4$, t(5) = 8.452, $p = 0.0004$; E2-I2, nucleus, $n = 6$, synapse, $n = 4$,

t(5.004) = 8.159, $p = 0.0004$; I5, nucleus, $n = 6$, synapse, $n = 4$, t(5.022) = 7.948, $p = 0.0005$; E1-E2-E3, nucleus, $n = 12$, synapse, $n = 9$, t(8.01) = 4.774, $p = 0.0014$; E8-E12, nucleus, $n = 12$, synapse, $n = 9$, t(8,083) = 3.834, $p = 0.0049$). ***$p < 0.005$, ****$p < 0.0001$. Error bars represent S.E.M. **e** RT-qPCR of *Gas5* variant ENSMUST00000162558.7 in the nucleus and synapse fractions of ILPFC samples. Statistical significance was determined using two-tailed unpaired Student's *t* test on independent biological replicates (Gas5, nucleus, $n = 6$, synapse, $n = 9$, t(8.05) = 3.453, $p = 0.0086$; Meg3, nucleus, $n = 8$, synapse, $n = 6$, t(7) = 9.763, $p < 0.0001$). **$p < 0.01$, ****$p < 0.0001$. Error bars represent S.E.M. **f** Representative image showing the localized expression of the *Gas5* variant in primary cortical neurons ($n = 8$ fields of views). Arrowheads show synaptic localization. Scale bar, 20 μm. Red, *Gas5*; blue, DAPI; green, MAP2 protein. The boxed region is enlarged in the inserts. Scale bar, 5 μm.

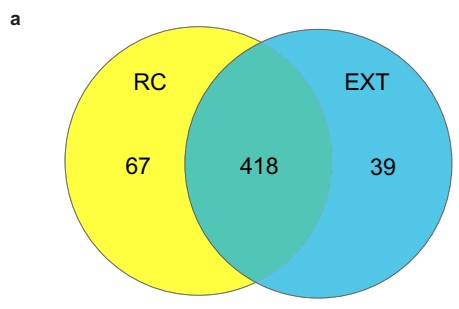

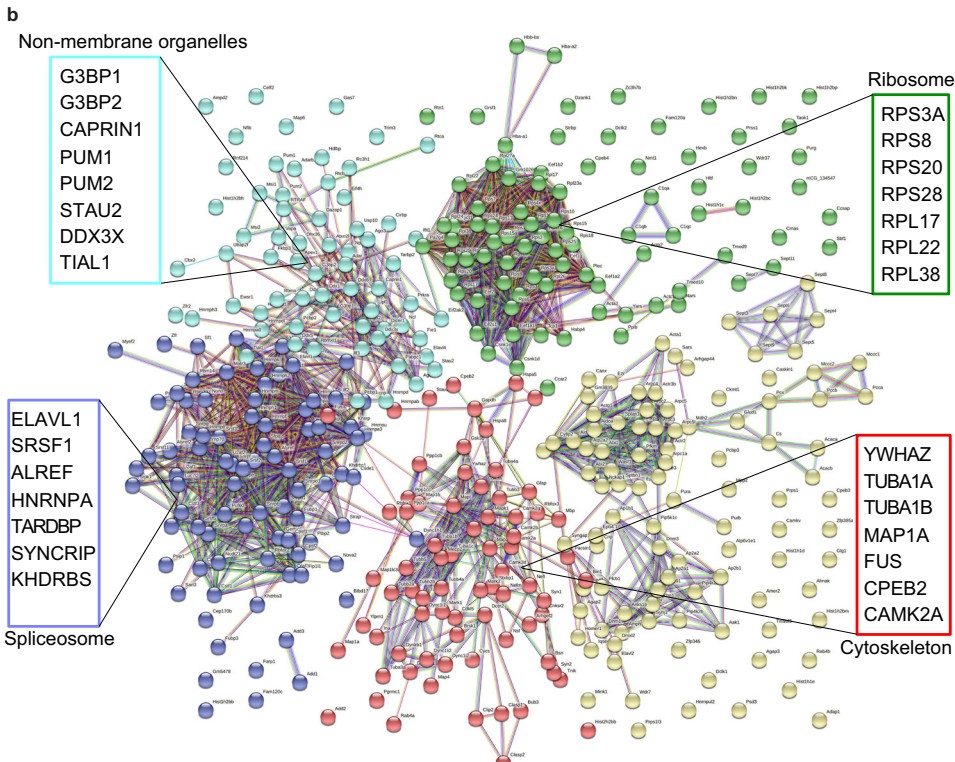

**Fig. 3 | *Gas5* interacts with proteins involved in translation, and RNA localization, as well as RNA granules. a** Venn diagram showing the number of unique and common proteins that bind to *Gas5* in both the RC and EXT groups. **b** STRING network analysis of the proteins bound to *Gas5* in both the RC and EXT groups. Only interactions with a STRING score ≥0.7 are shown. Evidence of interaction is represented by the distance between the nodes, with more tightly packed nodes having a higher STRING score. In both groups, membraneless organelles, ribosomes, cytoskeleton, and spliceosome clusters that contain tightly packed nodes are depicted. $n = 3$ independent biological replicates per group.

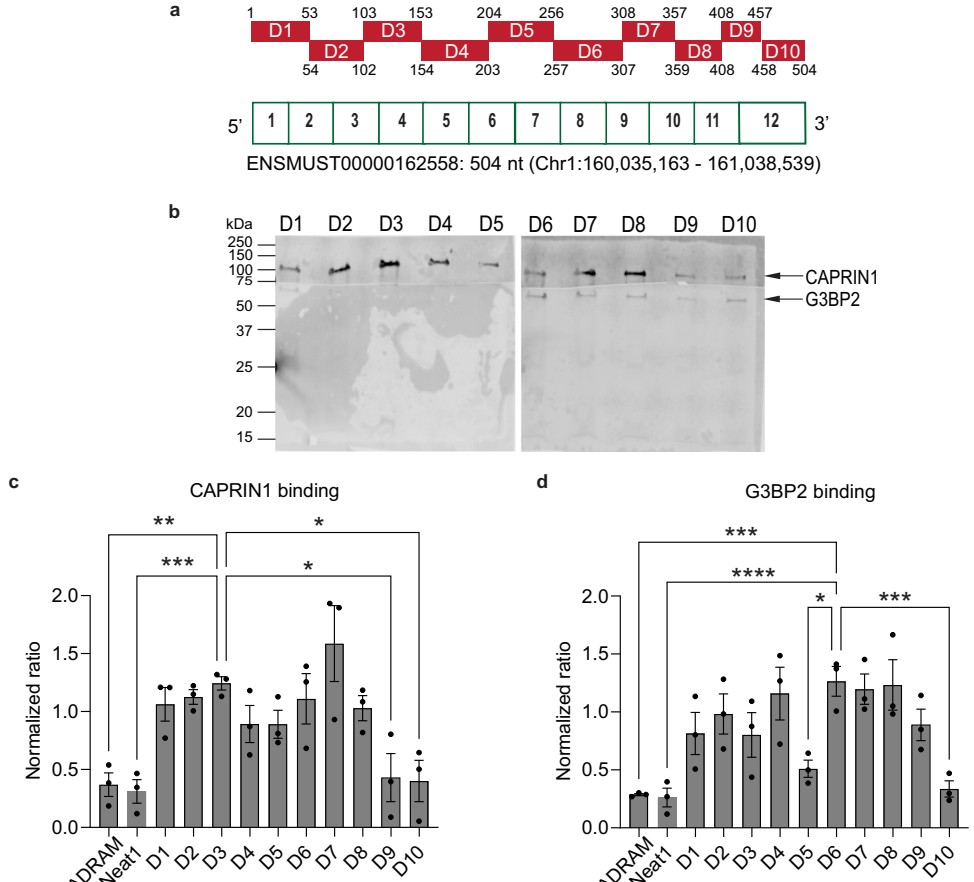

**Fig. 4 | *Gas5* interacts with CAPRIN1 and G3BP2 using distinct structural modules. a** Schematic showing deleted regions (D1–D10) of *Gas5* RNA fragments used for the in vitro biotinylated RNA pull-down assay to identify the CAPRIN1- and G3BP2-binding regions along the *Gas5* variant. The start and end position of the deleted regions are indicated either above or bottom of each red bars. **b** Western Blots displaying CAPRIN1 and G3BP2 protein expression after incubating different fragments of in-vitro transcribed *Gas5* with ILPFC protein extracts based on $n = 3$ experimental replicates. Band intensity values of **c** CAPRIN1 and **d** G3BP2 are normalized to their undeleted full-length control and lncRNAs, ADRAM and *Neat1*, was used as negative controls, one-way ANOVA for CAPRIN1 ($F_{(11,24)} = 5.99$, $p = 0.0001$; Dunnett's post hoc tests: D3 versus D9, $p = 0.0163$; D3 versus D10, $p = 0.0123$; D3 versus ADRAM, $p = 0.009$; D3 versus *Neat1*, $p = 0.0049$) and G3BP2 ($F_{(11,24)} = 6.438$, $p < 0.0001$; Dunnett's post hoc tests: D6 versus D5, $p = 0.0128$; D6 versus D10, $p = 0.0017$; D6 versus ADRAM, $p = 0.001$; D6 versus *Neat1*, $p = 0.0007$). *$p < 0.05$, **$p < 0.01$, ***$p < 0.005$, ****$p < 0.001$. Error bars represent S.E.M.

G3BP2-containing RNA granules, and that CAPRIN1 is critically involved in the regulation of postsynaptic AMPAR insertion[26], we next examined the effect of localized *Gas5* knockdown on synaptic activity by measuring the amplitude and frequency of miniature excitatory postsynaptic currents (mEPSCs) in cultured cortical neurons under low KCl stimulation conditions. *Gas5* knockdown led to a significant increase in both mEPSC amplitude ($1.93 \pm 0.527$ pA, $p < 0.001$) and frequency ($0.63 \pm 0.23$ Hz, $p < 0.01$) (Fig. 5l–n), suggesting that *Gas5* governs intrinsic neuronal excitability through its effect on the trafficking and clustering of RNA granules at the synapse.

### Targeted knockdown of the synapse-enriched *Gas5* variant in the ILPFC impairs fear extinction memory

To assess the functional role of the *Gas5* variant at the synapse in fear extinction, we examined the effect of synapse-targeted CIRTS-*Gas5*-mediated *Gas5* knockdown on extinction memory. The knockdown efficiency in vivo was validated by RT-qPCR (Fig. 6a, b), which showed a selective reduction in *Gas5* in the synaptic compartment, with little effect on nuclear *Gas5* expression. We also measured the expression of other *Gas5* splice variants, revealing a modest but non-significant effect of CIRTS-*Gas5* on their expression (Supplementary Fig. 9a). In addition, as demonstrated using RNAScope, infusion of the CIRTS-*Gas5* construct reduced endogenous *Gas5* expression without

affecting neuronal viability as nuclear staining (DAPI) within the ILPFC was similar to that in neighboring brain regions (Supplementary Fig. 10). Following infusion of the CIRTS-*Gas5* or control virus into the ILPC (Fig. 6c), there was no effect of *Gas5* knockdown on within-session fear extinction learning (Fig. 6d) or the ability to express fear in the absence of fear extinction when tested 24 h after training (Fig. 6e). In contrast, targeted *Gas5* knockdown led to a significant impairment in fear extinction memory when tested in context B (Fig. 6e). To determine the effect of *Gas5* knockdown on the stability of the original fear memory, the mice were re-exposed to the original training context A, 24 h later. There was no significant difference in fear expression between RC control virus and RC CIRTS-*Gas5*-treated animals, either in context A or B, suggesting that *Gas5* knockdown did not alter the stability of the original fear memory (Fig. 6e and Supplementary Fig. 9b). Collectively, these data indicate that synapse-directed *Gas5* knockdown in the ILPFC selectively impairs fear extinction memory, without interfering with the original fear memory trace.

## Discussion

In this study, we have discovered a significant population of lncRNAs at the synapse that includes a localized isoform of *Gas5* that is required for fear extinction memory. This *Gas5* variant interacts with CAPRIN1 and G3BP2, key proteins involved in translation, RNA trafficking, RNA

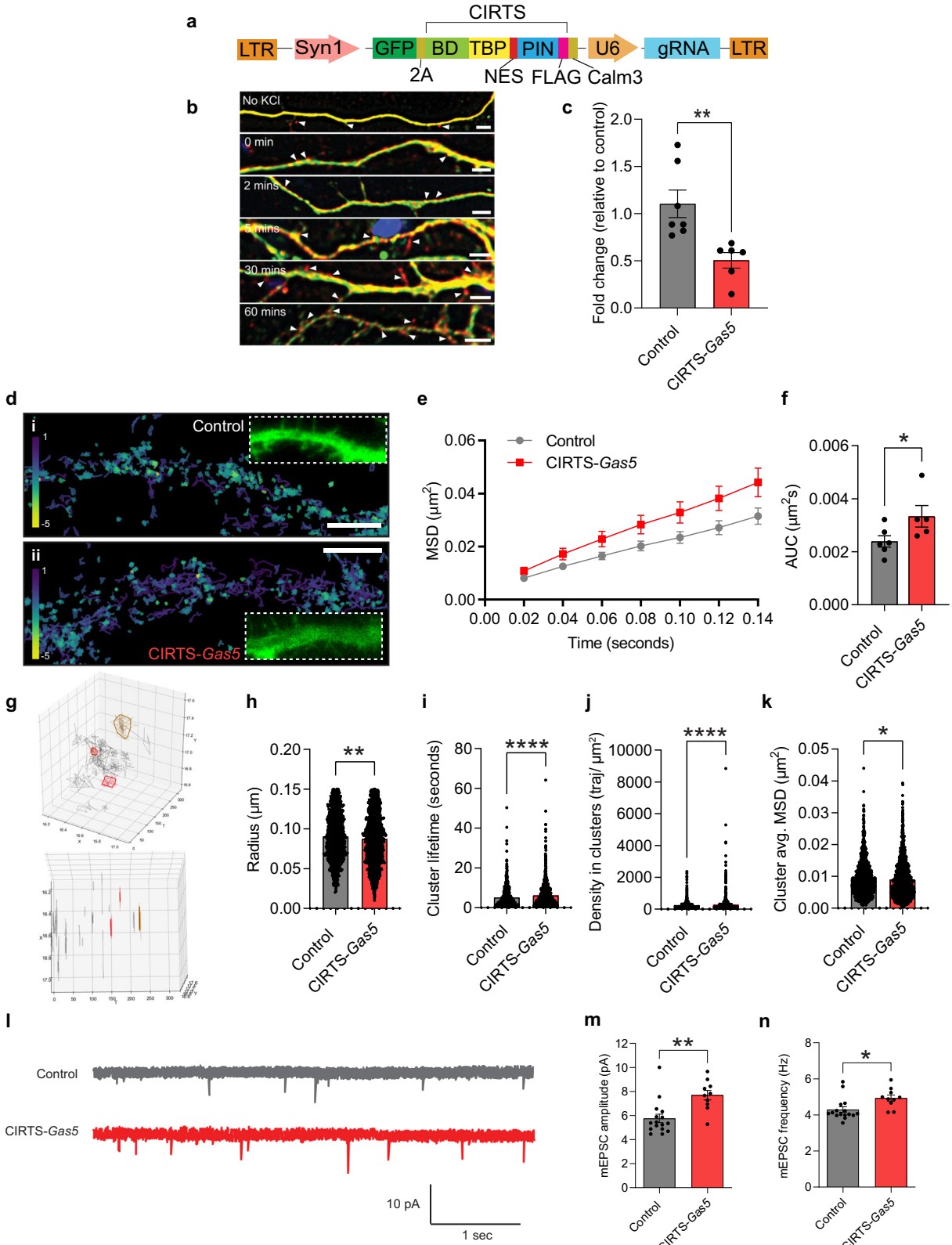

metabolism and RNA granule assembly at the synapse. An increase in intrinsic excitability and localized disassembly of *Gas5*-containing RNA granules following a reduction in *Gas5* at the synapse strongly supports a synapse-specific role for *Gas5* in the control of synaptic mechanisms underlying fear extinction memory. Together, these observations suggest a mechanism by which lncRNA activity at the synapse can influence the behavior of RNA condensates through the

binding of granule proteins. *Gas5* may therefore serve to coordinate the trafficking and clustering of RNA granules, and to organize the learning-induced activity of key proteins involved in local protein translation and subsequent memory formation (Fig. 7).

In agreement with this hypothesis, previous work has shown that lncRNAs can associate with ribonucleoproteins (RNPs) and other RNAs to form RNA condensates. For example, neuroLNC can regulate TDP-

**Fig. 5 | *Gas5* knockdown impairs the mobility and clustering of RNA granules and regulates intrinsic excitability. a** Schematic representation of the viral CIRTS knockdown construct. **b** expression of CIRTS along dendrites in primary cortical neurons. Time chased after 10 min of KCl induction is indicated. Arrowheads show CIRTS puncta (red) ($n = 8$ field of views). Scale bar = 5 μm. green, MAP2. **c** qRT-PCR performed on primary cortical neurons transduced with either control ($n = 7$ independent replicates) or CIRTS-*Gas5* ($n = 6$ independent replicates) (two-tailed unpaired Student's *t* test, t(9.329) = 3.584, $p = 0.0056$). **$p < 0.01$. Error bars represent S.E.M. **d** Representative primary cortical neurons dendritic region of interest illustrated from i) control and ii) a CIRTS-*Gas5* knockdown neuron. Calibration bar shows the $\log_{10}$ diffusion coefficient, yellow colors represent areas of lower diffusion. Scale bar, 2.5 μm. Graph displaying (**e**) the mean squared displacement (μm²) and (**f**) the area under the MSD curve (μm² s) for control ($n = 6$ independent replicates) and CIRTS-*Gas5* knockdown neurons ($n = 5$ independent replicates) (two-tailed unpaired Student's *t* test, $p = 0.0303$). *$p < 0.05$. Mean

±S.E.M. are plotted. **g** 3D representation of a region of interest highlights individual clusters (top), and their appearance in time (bottom). Spatiotemporal clusters were analyzed for their (**h**) radius (μm), (**i**) cluster lifetime (s), (**j**) density in clusters (traj/μm²) and (**k**) cluster average MSD (μm²) (Control, $n = 1244$ clusters; CIRTS-*Gas5*, $n = 1811$ clusters) (two-tailed unpaired Student's *t* test, radius, t(2760) = 3.042, $p = 0.0024$; cluster lifetime, t(2921) = 5.354, $p < 0.0001$; density in clusters, t(2950) = 5.596, $p < 0.0001$; cluster average, t(2724) = 1.967, $p = 0.0493$). *$p < 0.05$, **$p < 0.01$, ****$p < 0.0001$. Mean ± S.E.M. are plotted. **l** Traces of mEPSCs recorded in primary cortical neurons in the presence of control or CIRTS-*Gas5* knockdown. Plot showing (**m**) mEPSC amplitude and (**n**) mEPSC frequency of primary cortical neurons in the presence of control ($n = 16$ independent replicates) or CIRTS-*Gas5* knockdown ($n = 10$ independent replicates) (two-tailed unpaired Student's *t* test, amplitude, t(20.89) = 3.667, $p = 0.0014$; frequency, t(20.82) = 2.74, $p = 0.0123$). *$p < 0.05$, **$p < 0.01$. Error bars represent S.E.M.

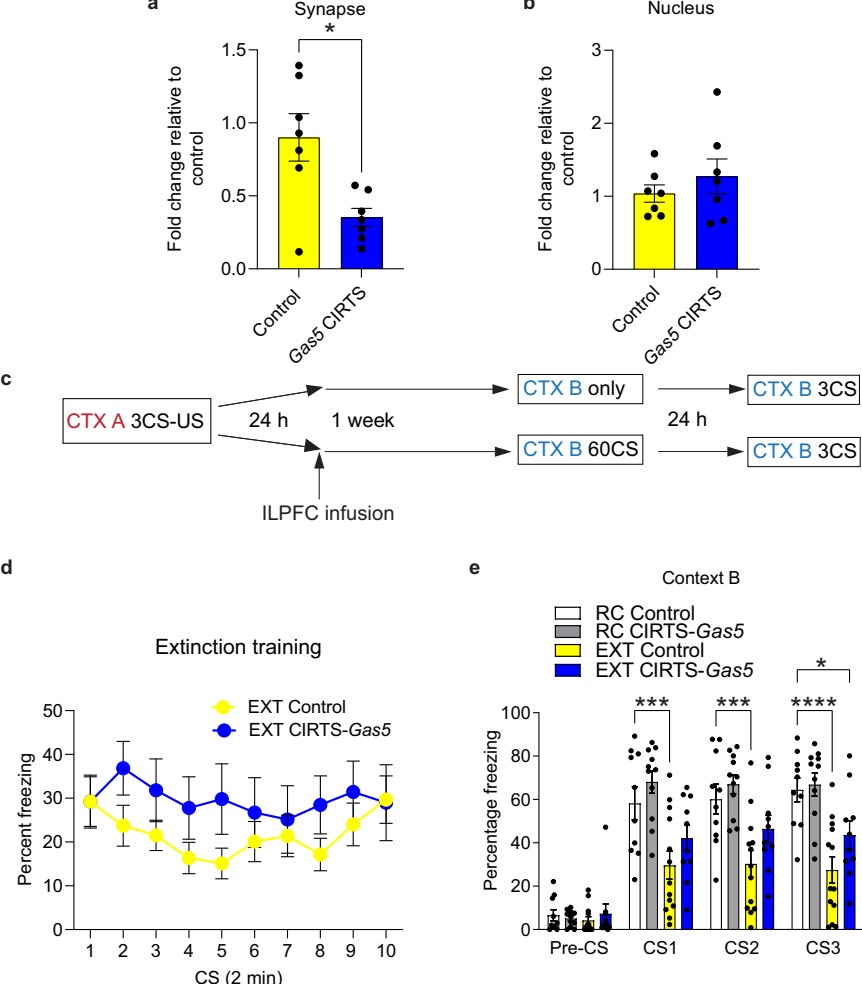

**Fig. 6 | Targeted *Gas5* knockdown impairs fear extinction memory.**
**a** Subcompartment-specific CIRTS-*Gas5*-mediated *Gas5* knockdown leads to a reduction in endogenous *Gas5* expression at the synapse ($n = 7$ independent biological replicates per group, two-tailed unpaired Student's *t* test, t(7.681) = 3.158, $p = 0.0142$) *$p < 0.05$, with (**b**) no effect on *Gas5* expression in the nucleus ($n = 7$ independent biological replicates per group, two-tailed unpaired Student's *t* test, t(8.77) = 0.8795, $p = 0.4026$). **c** Schematic of the behavioral protocol used to test the effect of *Gas5* knockdown in the ILPFC on fear extinction memory. CTX context, CS conditioned stimulus, US unconditioned stimulus. **d** There was no effect of *Gas5* knockdown on within-session performance during fear extinction training (EXT Control, $n = 13$ independent biological replicates per group, EXT CIRTS-*Gas5*, $n = 8$ independent biological replicates per group, two-way repeated measures ANOVA, $F$

$(1,19) = 1.447$, $p = 0.2438$). **e** There was no effect of *Gas5* knockdown on the ability to express fear memory (RC Control vs RC CIRTS-*Gas5*); however, mice treated with CIRTS-*Gas5* exhibit impaired fear extinction memory (RC Control, $n = 10$ independent biological replicates per group, RC CIRTS-*Gas5*, $n = 11$ independent biological replicates per group, EXT Control, $n = 13$ independent biological replicates per group, EXT CIRTS-*Gas5*, $n = 8$ independent biological replicates per group, two-way ANOVA, $F_{3,38} = 8.995$, $p = 0.0001$; Dunnett's post hoc tests: RC Control versus EXT Control, CS1 ***$p = 0.0007$, CS2 ***$p = 0.0004$, CS3 ****$p < 0.0001$; RC Control versus EXT CIRTS-*Gas5*, CS1 $p = 0.0994$, CS2 $p = 0.084$, CS3 *$p = 0.0158$; RC Control versus RC CIRTS-*Gas5*, CS1 $p = 0.4485$, CS2 $p = 0.7067$, CS3 $p = 0.9808$). Error bars represent S.E.M.

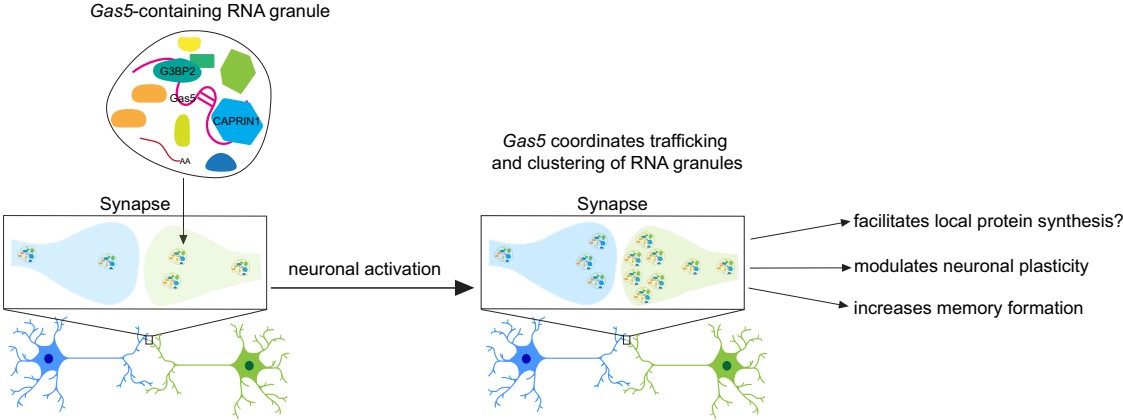

**Fig. 7 | Model of the proposed mechanism by which *Gas5* influences synaptic activity and the formation of fear extinction memory.** Extinction learning leads to the accumulation of the *Gas5* variant in the synaptic compartment, which then sequesters CAPRIN1 and G3BP2 containing RNA granules away from clustering, leading to an increase in local protein synthesis and tighter control over synaptic plasticity.

43 granule localization to the synapse[32,33]. Another study also suggested that CAPRIN1-containing granules are transported to dendrites and are important for long-term memory formation[26]. The formation of RNA granules is therefore dependent on the properties of RNA and protein, as well as the surrounding microenvironment[34,35]. We found that the "RNP complex" was the most abundant protein category in the RNA-protein network, with proteins involved in RNA granule formation being the most prominent. Our findings demonstrate that the synapse-enriched *Gas5* variant binds to the RNA granule proteins CAPRIN1 and G3BP2. These proteins are known to form a stable complex in response to stress[25], which serves to regulate condensate localization and dynamics. CAPRIN1-containing granules are also important for AMPAR loading and dendritic RNA localization in postsynaptic neurons[26]. Our results revealed that *Gas5* knockdown affects the mobility and trafficking of G3BP2-containing RNA granules, suggesting that *Gas5* acts as a scaffold to coordinate the clustering of RNA granules in an activity-dependent manner. It is therefore plausible that the destabilization of these "memory" granules represents a critical factor associated with impairments in the formation of long-term extinction memory.

The *Gas5* lncRNA is subject to a significant degree of alternative splicing, yielding many isoforms. Our findings indicate that the synaptic *Gas5* variant in the ILPFC is a mature isoform that is functionally involved in fear extinction memory. It is interesting to note that a different splice variant of *Gas5* detected in the nucleus accumbens plays a crucial role in motivated behavior[22], further suggesting a role for alternative splicing in brain-region-specific expression of different *Gas5* lncRNA variants. Most *Gas5* exons are flanked by small nucleolar RNA (snoRNAs) found within their introns, a unique feature found in sno-lncRNAs[36]. We found that intron-retained *Gas5* transcripts are generally enriched in the nucleus, suggesting that these transcripts have not yet had their introns removed[37] and may therefore represent processing intermediates, an observation that may also explain why synaptic lncRNAs are shorter and contain more exons. It is also possible that these lncRNAs contain UTRs that are crucial for their localization in different subcellular compartments[38] although this remains to be determined. As most *Gas5* transcripts have at least one snoRNA, they may also function as guides for RNA modification in the nucleus. Indeed, intron-retained, poorly spliced lncRNAs are more commonly found in the nucleus[23,39]. In addition to a multifunctional role for *Gas5* as a sno-lncRNA, it also remains possible that the *Gas5* lncRNA is capable of both coding and coding-independent functions, similar to those described for coding and noncoding RNAs[40,41]. In support of this idea, the human ortholog of *Gas5* is just one of many lncRNAs that contain short open reading frames (sORFs), which may yield micropeptides that localize to different subcellular compartments and are involved in diverse cellular functions[42]. Future studies will investigate whether sORFs derived from the *Gas5* locus are translated and functionally active in the adult brain.

Finally, many of the identified synapse-enriched lncRNAs were unannotated or labeled as pseudogenes with no function described. The term "pseudogene" was first used to categorize non-functional duplicated genes[43]. However, recent studies have shown that duplicated pseudogenes may be lncRNAs[44]. Indeed, some pseudogene-derived lncRNAs are induced in an activity-dependent manner and are required to maintain synaptic plasticity[45]. Interestingly, we found that many pseudogenic lncRNAs contain LINE and SINE elements, suggesting that these elements may be associated with their function. Recent studies have also shown that SINE- and LINE- containing lncRNAs can direct local translation of target genes and are often isoform-specific[46,47]. Given that lncRNAs have been shown to contain various structural modules that can also serve as decoys for microRNAs[48], as guides for axon regeneration[49] or as scaffolds for ribonucleoprotein localization[50,51], and the fact that dual functioning lncRNAs are increasingly becoming the norm rather than the exception[9,52,53], future studies should aim to dissect the structure-function relationship of each module, determine its conservation, and identify its role in neuronal function.

The work presented here leverages the lncRNA capture-seq approach, specifically designed to enhance the identification of low-abundance lncRNAs that are often overlooked by broader techniques like standard RNA-seq. However, while capture-seq is efficient, it has inherent limitations. One notable limitation is its susceptibility to coverage bias due to its reliance on current, incomplete lncRNA annotations for probe targeting. This limitation suggests that certain lncRNAs, including specific isoforms or entirely new entities, could go unnoticed. The recently released databases based on the new version of the mouse genome (mm39), which identifies more isoforms, including additional *Gas5* isoforms not mentioned in our study, highlights this limitation. Moreover, technical obstacles such as input requirements and the obtainable RNA amount from the synaptic compartment constrain our ability to achieve cell-type resolution. Given the aforementioned incomplete lncRNA annotations, it is possible that other lncRNAs beyond those reported in this work might be present at the synapse during fear extinction learning. Increasing the sample input and further refinement of the lncRNA set targeted by the probes are needed to capture the full spectrum of synapse-enriched lncRNAs in the ILPFC. We have functionally characterized a single lncRNA that is required for the mobility of one type of granule in the dendrite; however, this mechanism may be true for other lncRNAs or

classes of RNA granules. Further work is therefore required to explore the functional and mechanistic roles of individual synaptic lncRNAs and their interacting partners in different activity-dependent conditions and in other learning paradigms. In addition, the *Gas5* CIRTS gRNA may exhibit off-target effects, which could be partially mitigated with the use of a second gRNA along with whole transcriptome sequencing to assess the integrity of other transcripts following targeted *Gas5* knockdown.

In summary, we have discovered a significant number of alternatively spliced lncRNAs, including those that are enriched at the synapse. Specifically, we have revealed a localized isoform of the lncRNA *Gas5* that regulates the trafficking and clustering of RNA granules in dendrites, influences intrinsic neuronal excitability and drives the formation of fear extinction memory. These findings identify a new mechanism of fear extinction that involves the dynamic interaction between local lncRNA activity and the coordination of RNA condensates at the synapse.

## Methods

### Animals

Male C57BL/6 mice (10–14 weeks old) were housed two per cage, maintained on a 12 h light/dark time schedule at 18–24 °C and a relative humidity between 30–70% and allowed free access to food and water. All testing took place during the light phase in red-light-illuminated testing rooms following protocols approved by the Institutional Animal Care and Use Committee of the University of Queensland.

### Plasmid construction

The pFsy(1.1)GW lentiviral expression vector (Addgene #27232), containing the synapsin 1 promoter, was used to make the Syn1-Cirts-Calm3 construct. The CIRTs cassette was PCR amplified and cloned into the AgeI and XbaI sites. A NheI site was generated upstream of XbaI using PCR primer. The Calm3 dendritic localization sequence was PCR amplified from psiCheck2-Calm3 and cloned into the NheI and XbaI sites. The U6-*Gas5* gRNA and U6-scrambled control gRNA was PCR amplified and inserted in the XbaI site. GFP was PCR amplified and cloned into the AgeI site along with a 2A peptide signal. G3BP2-mEos3.2 was constructed by fusing mEos3.2 to the C-terminus of G3BP2. The fragment was PCR amplified and inserted into the AgeI and XbaI sites of the pFsy(1.1)GW lentiviral vector.

### Tissue culture

Cortical tissue was isolated from embryonic day 16–17 C57BL/6 embryos and primary cortical neurons were isolated after removing the skull and meninges with fine-tipped tweezers and brain tissue dissociated. Cells were then mixed with Neurobasal medium (Gibco) containing 5% fetal bovine serum (FBS), B27 supplement (Gibco), GlutaMAX (Gibco) and 1% penicillin-streptomycin (Gibco) and made homogenous with gentle pipetting. They were then passed through a 40 μm cell strainer (BD Falcon) and plated onto culture dishes coated with poly-L-ornithine (Sigma). HEK293T cells were maintained in a medium containing Dulbecco's modified Eagle's medium (DMEM) and high glucose (Gibco) with 5% FBS and 1% penicillin-streptomycin.

### Synaptosome preparation

The preparation of synaptosomes was carried out as previously described[54]. Briefly, the prefrontal cortices of four mice were homogenized in homogenizing buffer (250 mM sucrose, 5 mM Tris-HCl (pH7.5), 5 mM DTT, 0.1 mM RNaseOUT (Invitrogen)) with 10–14 strokes using a Teflon-glass tissue grinder. The homogenate was centrifuged at $1000 \times g$ for 10 min at 4 °C. The nucleus-enriched pellet was kept for subsequent analysis. The supernatant was directly applied onto a discontinuous Percoll gradient ranging from 0% up to 23% Percoll (GE Healthcare) and centrifuged at $31,000 \times g$ for 5 min to isolate the synaptosome fraction. The purity of synaptosomes was assessed by

Western blot using PSD-95 (1:2000, Abcam), Synaptophysin (1:20,000, Abcam) and HDAC2 (1:1000, Cell Signaling) markers.

### RNA extraction

Cultured cells, synaptosomes and tissues were homogenized using a Dounce tissue grinder in NucleoZOL (Macherey-Nagel) supplemented with 5 mM of DTT (Thermo Scientific) and 0.1 mM of RNaseOUT (Invitrogen). Samples were centrifuged for 15 min at $12,000 \times g$. Supernatant containing the total RNA was precipitated with 100% ethanol and purified using the RNA Clean and Concentrator Kits (Zymo Research), in-column DNase digested and extracted according to the manufacturer's instructions. The concentration of RNA was measured using a nanophotometer (IMPLEN) or Qubit fluorometer (Thermo Fisher Scientific).

### RT−qPCR

1 μg of RNA was used for cDNA synthesis using the QuantiTect Reverse Transcription Kit according to the manufacturer's protocol (Qiagen). Quantitative PCR was carried out on a RotorGeneQ (Qiagen) real-time PCR cycler with SensiFAST SYBR master mix (Bioline) using primers for target genes. All transcript levels were normalized to 18S rRNA using the ΔΔCT method and each PCR reaction was run in duplicate for each sample and repeated at least twice.

### Western blot

Homogenized issue and synaptosomes were fractionated in NP40 cell lysis buffer (Thermo Fisher Scientific). Briefly, samples were incubated on ice in lysis buffer for 30 min and then centrifuged at $17,968 \times g$ for 10 min at 4 °C. The supernatant was transferred to a new tube and protein concentration was measured using the Bradford assay (Sigma) on a nanophotometer (IMPLEN). Protein was diluted in Laemmli sample buffer with 5% 2-mercaptoethanol (Sigma-Aldrich) and denatured for 5 min at 95 °C. Gels were run and proteins transferred onto PVDF membrane (BioRad). The membrane was blocked with Odyssey Blocking Buffer (Li-Cor) for 1 h at room temperature and incubated with primary antibody overnight at 4 °C. The primary antibodies used were anti-CAPRIN1 (1:2000, Proteintech), G3BP2 (1:2000, Abcam), PSD95 (1:2000, Abcam), and beta-actin (1:1000, Cell Signaling). The membrane was washed with phosphate buffered saline containing 0.2% Triton X-100 (PBST) (3x), incubated for 1 h with IRDye 800CW secondary antibody (Li-COR) at 1:15,000 in PBST, and washed in PBST for 10 min (3x). Blot images were acquired using an Odyssey Fc system (Li-COR).

### Lentiviral production

Plasmid was co-transfected with pMD2.G (Addgene #12259), pRSV-Rev (Addgene #12253) and pMDLg/pRRE (Addgene 12251) into HEK293T cells at approximately 80% confluence using Lipofectamine 3000 transfection reagent (Thermo Fisher Scientific). 4 h later, sodium butyrate (Sigma) was added to stimulate viral production. After 2 days' incubation at 37 °C and 5% $CO_2$, the virus was collected by ultracentrifugation. The titer was measured using a Lenti-X qRT-PCR titration kit (Clontech).

### Lentiviral infusion

Lentivirus was prepared as previously described[55,56]. Double cannulae (PlasticsOne) were implanted in the anterior posterior plane, along the midline into the infralimbic prefrontal cortex (ILPFC), a minimum of 7 days prior to viral infusions. Injection coordinates were centered at +1.85 mm in the anterior posterior (AP) plane and −2.5 mm in the dorsal-ventral (DV) plane. A total of 2 μL of lentivirus was introduced via two injections, delivered at a rate of 0.1 uL/min, 48 h apart. 24-h prior to lentiviral infusions animals were fear conditioned as described below. One-week after lentiviral infusions mice were extinction trained.

## Behavioral training and analysis

According to our previously published protocol[55,56], two contexts (A and B) were used for all behavioral fear testing. Both conditioning chambers (Coulbourn Instruments) had two transparent walls and two stainless steel walls with a steel grid floors (3.2 mm in diameter, 8 mm centers); however, the grid floors in context B were covered by flat white plastic transparent surface. This surface was used to minimize context generalization. Digital cameras were mounted in the ceiling of each chamber and connected via a quad processor for automated scoring in a freezing measurement program (FreezeFrame). Fear conditioning (context A) was performed with a spray of lemon-alcohol (5% lemon and 10% alcohol). The fear-conditioning protocol started with a 120 s exposure to context A, which was then followed by three pairings of a 120 s, 80 dB white noise conditioned stimulus (CS) co-terminating with a 1 s, 0.7 mA foot shock (US). The trials were separated by a 120 s intertrial interval (ITI). The mice were then matched into equivalent treatment groups (CIRTS-*Gas5* virus or control virus) based on their freezing scores during the 3rd CS-pairing. Mice that exhibited less than 30% freezing during the 3rd CS-US pairing were excluded from further analysis. For extinction (context B) which was performed with a spray of vinegar, mice were again allowed to acclimate for 120 s and then extinction trained with 60 non-reinforced 120 s CS presentations with a 5 s ITI between CS exposures (60CS training protocol). For the behavioral control experiments, animals did not receive the CS (retention control - RC). Memory was tested by returning the animals to context B (24 h later) and presented with a 3 120 s CS with 120 s ITI. Memory was calculated as the percentage of time spent freezing during the tests. After training, viral spread and knockdown were assessed by RNAscope and immunohistochemistry.

## Immunofluorescence

Primary cortical neurons were fixed in 10% neutral buffered formalin solution (Sigma-Aldrich) at room temperature for 30 min, after which they were washed in PBS (3x) and incubated with 4% goat serum in PBST at room temperature for 1 h. Cells were then incubated with primary antibody at 4 °C overnight, washed with PBST (3x), and incubated with secondary antibody at room temperature for 1 h. Finally, cells were washed in PBS (3x), stained with DAPI and mounted on Superfrost Plus microscope slides (Thermo Fisher Scientific) with ProLong Gold Anti-fade Mountant (Thermo Fisher Scientific). Primary antibodies used were anti-MAP2 (1:2000, Abcam) and anti-GFP (1:2000, Abcam). Secondary antibodies were anti-chicken Alexa Fluor 488 (1:2000, Thermo Fisher Scientific), anti-rabbit Alexa Fluor 546 (1:500, Thermo Fisher Scientific) and anti-rabbit Alexa Fluor 647 (1:1000, Thermo Fisher Scientific). Neurons were imaged on an Axio Imager Z1 upright fluorescence microscope (Carl Zeiss) fitted with an Axiocam MRm camera (Carl Zeiss) and a 40X/0.75 NA Plan-Apochromat objective.

## RNAScope

Primary neurons were processed based on the manufacturer's instructions for the RNAScope RED assay (Advanced Cell Diagnostics) with few modifications and combined with an immunofluorescence protocol. First, samples were subjected to protease treatment, probe hybridization (RNAscope probe BA-Mm-*Gas5*-tv224-E8E9 ACD ADV833211), amplification and signal development. Blocking buffer containing 4% goat serum in PBST was added and incubated at room temperature for 1 h. Samples were then stained with primary and secondary antibodies (as described in the immunofluorescence protocol above). The amplified signal was detected using the cyanine 5 channel. The co-localized *Gas5* puncta in the nucleus and dendrites were quantified using Imaris software version 10.0.1 (Oxford Instruments).

For RNAScope on tissue sections, animals were perfused with 4% paraformaldehyde and brains were collected in 30% sucrose prior to slicing. Sectioning at 14 μm was performed using a Zeiss Microm

HM560 cryostat and sections were mounted on SuperFrost Plus slides (Thermo Fisher Scientific). The slides were then baked at 60 °C for 30 min, washed and incubated with RNAscope hydrogen peroxide solution for 10 min at room temperature. Slides were incubated with target retrieval solution for 5 min at 95 °C, followed by Protease Plus for 30 min at 40 °C. Sections were processed based on the manufacturer's instructions for the RNAScope RED assay. After the chromogen development step, sections were incubated for 1 h in blocking buffer and incubated with primary antibody at 4 °C overnight. Slices were washed with PBST (3x), after which secondary antibodies were added. Slices were incubated at room temperature for 1 h, washed 3 times with PBST and incubated with 4′,6-diamidino-2-phenylindole (DAPI) (Thermo Fisher Scientific) for 10 min at room temperature. Coverslips were applied with Vectamount Permanent Mounting Medium (ACD). Sections were imaged on a spinning-disk confocal system (Marianas; 3I, Inc.) consisting of a Axio Observer Z1 (Carl Zeiss) equipped with a CSU-W1 spinning-disk head (Yokogawa Corporation of America), ORCA-Flash4.0 v2 sCMOS camera (Hamamatsu Photonics), 20×0.8 NA PlanApo and 40×1.2 NA C-Apo objectives. Image acquisition was performed using SlideBook version 6.0 (3I, Inc).

## Image acquisition and analysis

Image acquisition was performed using the ZEN 2012 software (Carl Zeiss). Images were analyzed using ImageJ version 1.54 f and figures were constructed using the FigureJ plugin version 1.36[57].

## lncRNA capture sequencing

mPFC tissue was processed and nuclear fraction and synaptosome fraction were collected from pooled samples reflecting a total 12 RC and 4 EXT trained adult mice and total RNA was isolated as described above. 100–500 ng of rRNA depleted total synaptosome RNAs were used for library construction. The same amount of input was used for the pooled nucleus-derived RNAs, reflecting a total of 12 RC control and 12 EXT trained mice. cDNA libraries were generated with random primers using the NEBNext Ultra II RNA Library Prep Kit for Illumina (NEB) according to the manufacturer's protocol. At least 1 μg of cDNA was used for subsequent capture. A custom-designed panel of 190,689 probes (Roche) targeting 28,228 known and predicted mouse lncRNAs[20], and spanning 117,203 target regions. This panel, previously developed to improve the annotation of brain-enriched lncRNA, was used to capture amplified cDNA. Notably, some target regions are covered by multiple probes, explaining the discrepancy between the number of probes and the number of target regions. The capture procedure was performed using the SeqCap EZ Hybridization and Wash Kit (Roche) and SeqCap EZ Accessory kit (Roche) according to the manufacturer's instructions. Captured libraries were sequenced on an Illumina HiSeq 4000 platform with 150-bp paired-end reads (Genewiz). The sequencing depth for synaptosome samples was 56 million reads (8.4 Gbp) to 74 million reads (11.1 Gbp), with an average of 65.3 million reads, while the sequencing depth for the nucleus samples was 51 million reads (7.7 Gbp) to 128 million reads (19.2 Gbp), with an average of 82.4 million reads.

## Synaptosome RNA sequencing

Synaptosomes were collected from behaviorally trained adult mice, and total RNA was extracted as described above. 100–500 ng of total synaptosome RNA was used for library construction. cDNAs were generated using the SMARTer® Stranded Total RNA-Seq Kit v2 – Pico Input Mammalian (Takara). RNA-seq libraries were sequenced on an Illumina HiSeq 4000 platform with 150-bp paired-end reads (Genewiz). The sequencing depth for the synaptosome RC samples ranged from 37 million reads (5.5 Gbp) to 53 million reads (7.9 Gbp), averaging at 44.4 million reads. In contrast, the sequencing depth for the synaptosome EXT samples ranged from 47 million reads (7.1 Gbp) to 76 million reads (11.4 Gbp), with an average of 63.1 million reads.

## Sequencing data analysis

Cutadapt[58] (version 1.17, https://cutadapt.readthedocs.io/en/stable/) was used to trim low-quality nucleotides (Phred quality lower than 20) and Illumina adapter sequences at the 3' end of each read for lncRNA capture sequencing data. Processed reads were aligned to the mouse reference genome (mm10) using HISAT2 (version 2.1.0)[59]. SAMtools (version 1.8)[60] was then used to convert "SAM" files to "BAM" files, remove duplicate reads, and sort and index the "BAM" files. To avoid the artefact signals potentially introduced by misalignments, we only kept properly paired-end aligned reads with a mapping quality of at least 20 for downstream analyses.

For capture-seq, three rounds of StringTie (version 2.1.4)[61] were applied to i) perform reference-guided transcriptome assembly by supplying the GENCODE annotation file (V25) with the "-G" option for each sample, ii) generate a non-redundant set of transcripts using the StringTie merge mode, and iii) quantify the transcript-level expression for each sample, with the option of "-e -G merged.gtf". Known protein-coding transcripts (with the GENCODE biotype as "protein_coding") were removed from the StringTie results. We assessed the complexity of the transcriptome in both nuclear and synaptic compartments based on the total number of transcripts and the number of isoforms per gene. Because the level of transcriptome complexity is comparable between both compartments (Supplementary Fig. 11), we therefore used Ballgown (version 2.22.0)[62] to conduct transcript-level differential expression analysis. Alternative splicing analysis was performed using SUPPA2 (version 2.3, https://github.com/comprna/SUPPA)[63]. Differential expression of *Gas5* isoforms was created using a web-based visualization tool, IsoVis (version 1.1.1). Annotation reference and a file containing count data for each sample were used as input. IsoVis is available at https://isomix.org/.

For RNA-seq, the same pipeline prior to StringTie (version 2.1.4)[61] analysis was conducted as described above. The gene annotation file ("merged.gtf") used in capture-seq data analysis were supplied to StringTie (version 2.1.4) to quantify the transcript-level expression for each sample, with the option of "-e -G merged.gtf", and generated the normalized abundance data (FPKM) for each transcript. Ballgown (version 2.22.0)[62] was then used to conduct transcript-level differential expression analysis between the RC and EXT group.

## RNA pull-down assay

*Gas5* variant, deleted fragments, *Neat* and ADRAM were amplified using the T7 promoter sequence on the 5' end of the forward primers. The deleted *Gas5*, *Neat* and ADRAM DNA fragments were synthesized by IDT. The PCR products were gel extracted using the Gel DNA Recovery Kit (Zymo Research) and in-vitro transcribed using the HiScribe T7 Quick High Yield RNA Synthesis Kit (NEB) according to the manufacturer's instructions. The transcribed RNA was purified using the RNA Clean and Concentrator Kits (Zymo Research). Biotinylation and pull-down were performed using the Pierce Magnetic RNA-Protein Pull-Down Kit (Thermo Fisher) according to the manufacturer's instructions, and the concentration of biotinylated RNA was measured using a nanophotometer (IMPLEN). To check for the formation of highly stable structure, 10 μl of mutant RNAs were denatured at 95 °C for 2 min, then transferred to ice for 1 min. 4 μl of ice-cold 5X RNA folding buffer (500 mM HEPES, pH 8.0; 500 mM NaCl) supplemented with RNAseOUT inhibitor (Invitrogen) were added and the RNA was then incubated for 15 min at 37 °C to allow secondary structure formation. 2 μl of 100 mM MgCl₂ (pre-warmed at 37 °C) was added and RNA was further incubated for 15 min at 37 °C to allow tertiary structure formation. RNAs were visualized on a 1% native agarose gel. Briefly, ILFPC samples were incubated for 30 min on ice in NP40 cell lysis buffer (Thermo Fisher) supplemented with halt protease inhibitor (Thermo Fisher) and RNAseOUT RNase inhibitor (Thermo Fisher), and then centrifuged at $17,968 \times g$ for 10 min at 4 °C. The supernatant was transferred to a new tube and protein concentration was measured

using the Bradford assay (Sigma) on a nanophotometer (IMPLEN). 1 μg of biotinylated RNAs were incubated with Streptavidin beads for 30 min. The RNA-conjugated beads were washed three times and incubated with 500 μg of total ILPFC proteins for 1 h with rotation. RNA-protein-containing beads were UV-crosslinked, washed three times and subjected to either Western blot or mass spectrometry. For Western blot, the band intensity was measured using ImageJ. The integrity of the mutant RNAs were also assessed after incubation with proteins as described above. 2 μl of Proteinase K (NEB) was added to the RNA-protein-beads mixture and incubated at room temperature for 10 min. The RNAs were purified using the RNA Clean and Concentrator Kits (Zymo Research) and visualized on a native agarose gel.

## RNA immunoprecipitation

FLAG-tagged G3BP2 and CAPRIN1-expressing primary cortical neurons were crosslinked with 0.1% formaldehyde for 10 min, and incubated for 30 min on ice in NP40 cell lysis buffer (Thermo Fisher) supplemented with halt protease inhibitor (Thermo Fisher) and RNaseOUT RNase inhibitor (Thermo Fisher), and then centrifuged at $17,968 \times g$ for 10 min at 4 °C. The supernatant was transferred to a new tube and protein concentration was measured using the Bradford assay (Sigma) on a nanophotometer (IMPLEN). 2 μg of IgG (Cell Signaling) or FLAG antibody (Sigma) was added to the pre-cleared lysate and incubated for 2 h with rotation. Protein G beads (Thermo Fisher) were added to each IP samples and incubate for another 1 h with rotation. Beads were washed three times, pelleted and resuspended in NucleoZOL (Macherey-Nagel). RNAs were purified and subjected to RT-qPCR as described in the previous section.

For the in-vivo RNA immunoprecipitation assay, ILPFC tissues from behavioral-trained animals were crosslinked with 0.1% formaldehyde for 10 min, and homogenized in NP40 cell lysis buffer (Thermo Fisher) supplemented with halt protease inhibitor (Thermo Fisher) and RNaseOUT RNase inhibitor (Thermo Fisher), and incubated for 30 min on ice. Samples were then centrifuged at $17,968 \times g$ for 10 min at 4 °C, and supernatant was transferred to a new tube and protein concentration was measured as described above. 2 ug of CAPRIN1 antibody (Proteintech) was added to the pre-cleared lysate and incubated and processed as described above.

## Nuclease cleavage assay

The nuclease cleavage assay was performed as previously described with slight modification[28]. Briefly, 1 μg of total ILPFC RNA, 250 ng of gRNA and an equal amount of in-vitro translated CIRTS were incubated in nuclease buffer (20 mM HEPES, pH 7.5, 150 mM NaCl, 1 mM DTT, 0.5 mM MnCl₂, 10% glycerol) for 2 h at 37 °C. CIRTS proteins were synthesized using the TNT Quick Coupled Transcription/Translation Systems (Promega). The reaction was quenched in stop buffer (proteinase K (NEB) and 60 mM EDTA) for 30 min at 37 °C. RNAs were then purified using the RNA Clean and Concentrator Kits (Zymo Research) and subjected to RT-qPCR.

## HPLC/MS MS/MS, mass spectrometry and protein identification

Magnetic affinity beads were covered with 40 μl of 40 ng/μl sequence grade trypsin in 50 mM ammonium bicarbonate pH8 buffer (Promega). The beads were placed in an incubator at 37 °C overnight. The trypsin solution was removed from each sample and placed in a clean Eppendorf tube. 200 μl of 5% formic acid/acetonitrile (3:1 (vol/vol) was added to each tube and incubated for 30 min at room temperature in a shaker. The supernatant was placed into the pre-cleaned Eppendorf tubes, together with the trypsin solution for each sample and dried down in a vacuum centrifuge.

For HPLC/MS MS/MS analysis, 15 μl of 1.0% (vol/vol) TFA in water was added to the tube, which was vortexed and/or incubated for 2 min in the sonication bath and transferred to an autosampler vial for analysis. Tryptic peptide extracts were analyzed by microflow HPLC/MS

MS/MS on an Eksigent, Ekspert nano LC400 uHPLC (SCIEX) coupled to a Triple TOF 6600 mass spectrometer (SCIEX) equipped with a micro Duo IonSpray, ion source. 5 μl of each extract was injected onto a 5 mm × 300 μm, C18 3 μm trap column (SGE) for 6 min at 10 μl/min. The trapped tryptic peptide extracts were then washed onto the analytical 300 μm × 150 mm Zorbax 300SB-C18 3.5 μm column (Agilent Technologies) at 3 μl/min and a column temperature of 45 °C. Linear gradients of 2–25% solvent B over 60 min at 3 μl/min flow rate, followed by a steeper gradient from 25% to 35% solvent B in 13 min, then 35% to 80% solvent B in 2 min, were used for peptide elution. The gradient was then returned to 2% solvent B for equilibration prior to the next sample injection. Solvent A consisted of 0.1% formic acid in water and solvent B contained 0.1% formic acid in acetonitrile. The micro ionspray voltage was set to 5500 V, declustering potential (DP) 80 V, curtain gas flow 25, nebulizer gas 1 (GS1) 15, heater gas 2 (GS2) 30 and interface heater at 150ºC. The mass spectrometer acquired 250 ms full-scan TOF-MS data followed by up to 30, 50 ms full scan product ion data, with a rolling collision energy, in an Information Dependent Acquisition (IDA) scan mode. Full scan TOFMS data were acquired over the mass range m/z 350–2000 and for product ion ms/ms, m/z 100–1500. Ions observed in the TOF-MS scan exceeding a threshold of 150 counts and a charge state of +2 to +5 were set to trigger the acquisition of product ion, ms/ms spectra of the resultant 30 most intense ions. The data were acquired and processed using Analyst version 1.7 software (SCIEX). Protein identification was carried out using Protein Pilot version 5.0 (SCIEX) for database searching.

### Proteomics data and GO analysis
A network analysis was carried out using the STRING protein query database version 11.5 for *Mus musculus* using the official gene-symbol (https://string-db.org)[64]. The confidence score cut-off was set as 0.7. The p-values were corrected for multiple testing within each category using the Benjamini-Hochberg procedure.

### Single-molecule localization microscopy and analysis
Primary cortical neurons were cultured and imaged on a glass bottom dishes (Cellvis) in low K⁺ imaging buffer as reported previously[65]. Cells were then transduced with G3BP2-mEos3.2 and CIRTS and images were acquired on an ELYRA PSI microscope equipped with a x100/1.46NA α Plan-Apochromat oil-immersion objective and an EMCCD camera. sptPALM was performed as reported[66]. Briefly, G3BP2-mEos3.2-expressing cells were photo-converted with a 405 nm laser and excited using a 561 nm laser. 16,000 frames were acquired at a rate of 50 Hz. All data were acquired using Meta-Morph Microscopy Automation and Image Analysis software version 7.7.8 (Molecular Devices) and further processed using PalmTracer software version 2.0.4.1778[67]. Movies were subsequently drift corrected using SharpVisu version 1.3[68] and analyzed by Nanoscale Spatiotemporal Index Clustering (NASTIC) version 1[30] analysis was compared by Welch's two sample t-test.

### Primary cortical neuron recordings
Whole-cell patch clamp experiments were performed from cortical neurons in culture at DIV 18–25. The neurons were pre-incubated with the recording solution (145 mM NaCl, 5.6 mM KCl, 2.2 mM CaCl₂, 0.5 mM MgCl₂, 5.6 mM D-glucose, 15 mM HEPES (pH7.4) and 1 μM tetrodotoxin) for 1 h prior to recording. To measure the excitatory currents, an internal solution (100 mM cesium gluconate, 0.2 mM EGTA, 5 mM MgCl₂, 2 mM ATP, 0.3 mM GTP, 40 mM HEPES (pH 7.2) was back-filled in glass micropipettes to an open resistance of 3–8 MΩs. mEPSCs were recorded by holding the cells at −70 mV. All signals were recorded with a Multiclamp700B amplifier and Digidata1440A ADC and collected at a bin rate of 10 kHz. Post hoc analysis of mEPSC events were analyzed by pClamp version 10.5 software and Matlab version R2021b. The average of mEPSC events for 300 s was analyzed and selected from each neuron, which had peak amplitudes of <-4pA, rise rate of >0.3 pA/ms and decay time constants between 1–12 ms.

### Statistical analysis
Statistical analyses were performed using GraphPad Prism version 9 unless otherwise specified. Welch's two sample t-test was performed when comparing two categories. When more than two groups were compared, one-way ANOVA followed by a Dunnett's multiple comparisons test was used. Results are mean n ±standard error of the mean (s.e.m.) unless otherwise stated. For sequencing analysis, the ballgown package was used to perform a parametric F-test, and the p-value was adjusted to account for multiple testing corrections, which was reported in the Q-value column in Supplementary Data 1, 2 and 3. For behavioral analysis, the data represent the mean ± s.e.m. percent freezing for each group. All behavioral data analysis was carried out using a two-way ANOVA for the data in Fig. 6 and Supplementary Fig. 9. A Dunnett's test was used for post hoc comparisons with the RC control group.

### Reporting summary
Further information on research design is available in the Nature Portfolio Reporting Summary linked to this article.

## Data availability
Protein network analysis was performed using the STRING protein query database version 11.5 (https://string-db.org). All sequencing data generated in this study have been deposited in the NCBI Gene Expression Omnibus database under the accession code GSE207149. Proteomics data generated in this study have been deposited to the ProteomeXchange Consortium via the PRIDE[69] partner repository with the dataset identifier PXD046479. All image data generated in this study are available at Figshare[70] (https://doi.org/10.6084/m9.figshare.24431452). The data supporting the findings of this study are available from the corresponding authors upon request. Source data for the figures and supplementary figures are provided as a Source Data file. Source data are provided with this paper.

## Code availability
The sequencing data analysis pipeline and associated custom PERL scripts are available on GitHub (https://github.com/Qiongyi/lncRNA_nucleus_synapse).

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

## Acknowledgements

We thank Dr. Alun Jones from the Mass Spectrometry Facility in the Institute of Molecular Bioscience at the University of Queensland for assistance with the proteomics experiments and Ms. Rowan Tweedale for manuscript editing. The authors gratefully acknowledge the Queensland Brain Institute Advanced Microscopy Facility for its support. We thank JB Sibarita (University of Bordeaux, France) for providing the PalmTracer software and Josie Gleeson for assistance with the IsoVis figure. The authors acknowledge grant support from the Brain and Behavioral Research Foundation (NARSAD Independent Investigator Award, T.W.B.), NIH R01MH109588 to TWB and RCS, NHMRC Ideas Grant (GNT2003414) to T.W.B. and Investigator Grant (GNT1196841) to M.B.C., NSFC 82001421 to X.L., the NIGMS R35 GM119840 to B.C.D., the DFG SFB870 and SPP1935 to M.A.K., and the ARC DP190100674 to F.A.M. E.L.Z, L.J.L. and S.U.M. are supported by a Westpac Future Scholarship and the University of Queensland. S.B. and B.S. are supported by a core research grant from the Science and Engineering Research Board from the government of India.

## Author contributions

W.L. conceived this project, together with T.W.B. and R.C.S., and led the development and optimization of the protocol, performed experiments, analyzed and interpreted data, generated figures, and wrote and edited the paper. Q.Z. performed all bioinformatics analyses, interpreted data, generated figures, and wrote and edited the paper. M.C. shared the lncRNA probe set. X.L. and W.W. assisted with capture-seq experiment. H.G., H.R., L.J.L., M.M., J.D., S.U.M., A.P. and E.L.Z. assisted with tissue collection, surgery, and behavioral experiments. S.R., C.H., B.C.D., S.M.E. and M.A.K. assisted with the CIRTS construct design. P.R.M. performed the behavioral experiments and analysis. A.B. performed sptPALM experiments and A.B. and R.S.G. analyzed the data. F.A.M. supervised the sptPALM analysis. S.B. and B.S. performed the electro-physiological experiments. All authors discussed the results and contributed to the manuscript and discussion.

## Competing interests

C.H. is a scientific founder and a member of the scientific advisory board of Accent Therapeutics Inc. and Inferna Green Inc. BCD is a founder and holds equity in Tornado Bio, Inc. The remaining authors declare no competing interests.
