## [Peer Review File · Nature Communications]

Fear extinction is regulated by the activity of long noncoding RNAs at the synapseEditorial Note: This manuscript has been previously reviewed at another journal that is not operating a transparent peer review scheme. This document only contains reviewer comments and rebuttal letters for versions considered at *Nature Communications*.

REVIEWER COMMENTS

Reviewer #4 (Remarks to the Author):

In this revision, the authors have done substantial amount of work to address the numerous reviewer's comments raised upon the previous submission. The manuscript has therefore been significantly improved. Though the majority of my prior comments have been addressed in the rebuttal, I have the following questions that require further clarifications.

1. It is not sufficiently clear what have been done in the capture-seq. The subtitle of "results" states "... lncRNAs are enriched at the synapse in the adult ILPFC" (line 98), which seems to suggest the profiling in a healthy control. However, in the following paragraph, it says the capture seq was done "on tissue derived from the adult mouse ILPFC of extinction trained mice" (lines 106-107). I am confused if the capture-seq was only done in fear extinction trained animals, or also carried out in control mice? Should a differential analysis of synapse capture-seq between control and extinction groups be a valuable approach to recognize candidate lncRNAs implicated in fear learning?

2. I appreciate the authors' effort to add RNAseq to the revision to complement the lncRNA profiling derived from the capture-seq in the initial approach. However, as the authors acknowledged (rebuttal letter page 17), none of the ten differential synaptic lncRNAs was significantly changed in the RNAseq analysis. This was highlighted as an advantage of using the targeted sequencing system. However, what was the sequencing depth of the RNAseq? I can not find this information in the manuscript, though I saw the read numbers of capture-seq were added to the revision. Would a deeper sequencing of RNAseq allow the authors to better compare the low-frequency events? Related, it does not appear the transcriptome RNAseq is added to the "methods" section. In the "lncRNA capture sequencing" part of "methods", it mentioned the NEBNext Ultra II RNA library prep kit for illumina was employed (line 785). Does that kit have two modules, one for polyA RNA isolation, and the other for rRNA depletion? If so, which one was applied in the study? This may affect the population of lncRNA that was collected.

3. I have not found any descriptive information on the seven supplementary tables in the manuscript. Though I can guess, such as Table S4 appears to be the list of capture-seq probes, I am not certain about all (e.g., what is Table S5?). Some rows in a few tables (e.g., Table S1, S2, S3) are highlighted in color that should be explained as well. It says in the manuscript that the capture-seq system employed 190,689 probes, but there are only ~117K in table S4. Should these be consistent?

Reviewer #5 (Remarks to the Author):

The revised manuscript by Liau and colleagues has improved and become more clear, however several major weaknesses remain. Also, due to the rather minimalistic description of the experimental procedures, it hard at times to evaluate the scientific validity or meaning of an experiment. The most important issues are described below:

1) Identification of the lncRNAs that are enriched in the synaptic compartment clearly constitute an important section of the manuscript. However, unfortunately the studies addressing this important question are poorly described, controlled and confusing.

a) Critical aspects of RNA sequencing-based studies are not covered in the manuscript. For example, how was the normalization done when comparing synaptic and nuclear compartments? Similarly, when RNAs from synaptic and nuclear compartments were compared by RT-qPCR, were the results normalized to 18S rRNA or PGK, as stated in the methods?

b) The lack of concordance of the standard bulk RNA-seq without enrichment for lncRNAs versus the results following capture-seq amplifies this concern. This lack of concordance could be due to inherent technical issues of enrichment protocols or normalization problems described above, which can affect both standard and enriched RNA-seq studies. This is particularly a concern since the studied lncRNAs (which didn't show concordant expression patterns) were those having the highest expression levels among the lncRNAs in the subset of interest and thus, were unlikely to be under-represented in standard bulk RNA-seq due to inefficient sampling (which can happen with transcripts with very low copy numbers).

2) Another major section of the manuscript deals with defining the protein interactome of a GAS5 isoform that the authors believe is enriched in the synaptic compartment.

a) These studies are unfortunately performed using in vitro-transcribed RNAs incubated in cellular extracts. As RNAs fold and obtain their protein interactome co-transcriptionally, studies using in vitro-assembled systems are known to be highly prone to artifacts. The authors have attempted RIP studies to capture the interaction of their selected proteins with GAS5, however, there is no experimental details available about these studies. Whether the samples were crosslinked prior to lysis is not indicated and it seems that no RNA controls were used. In the absence of RNA pull down of endogenous GAS5, which has not been performed, RIP studies must be performed in a highly controlled manner to define the stoichiometry of the RNA-protein interactions described. Studies that merely show the presence of an interaction between two macromolecules can be misleading, since at some level, many abundant RNAs interact with many abundant proteins at low stoichiometry (e.g. 5% of the copies of the relevant isoform of GAS5 may interact with a certain protein). These interactions can be captured by RIP-seq in the absence of appropriate controls, but they don't carry functional significance.

b) As abundant RNA-binding proteins such as splicing factors and ribosomal factors bind any RNA to some extent, especially in an in vitro-assembled system such as the one used in the current study, ascribing functional significance to these likely fortuitous, concentration-driven interactions must be avoided (e.g. lines 193-208).

c) The use of lncRNA ADRAM is certainly a step in the right direction. However, it's not sufficient and additional negative control lncRNAs (not snRNAs or tRNAs) should be used or even better, existing RNA pull down studies, many of which are available in GEO and SRA, can be queried for the pull down of Caprin1 and G3bp2 to determine their potential for forming nonspecific RNA interactions.

d) Lines 221-225: deletion mutagenesis studies must be validated with studies to prove that all deletion mutants are present in the extracts at the same level and that these mutations don't generate highly stable structures that may artifactually result in altered interactions with proteins.

3) The functional studies using CIRT5 is certainly innovative and exciting. However, there are a number of issues with these studies.

a) Unfortunately with only a single functional gRNA, off-target effects can't be ruled out, reducing the scientific rigor of the shown data.

b) An important missing detail is whether the "scrambled" control RNA used in CIRT5 studies is still a gRNA (with an intact RNA hairpin), or if it lacks the ability to associate with the hairpin binding protein in the CIRT5 machinery.

c) The studies in Fig. 5, while interesting, lack a positive control. Since the impact of changes in GAS5 expression looks relatively modest, it would have been helpful to include a positive control (knock down of a known regulator of trafficking/assembly of G3bp2 containing condensates) to determine the biological significance of the observed changes.

Reviewer #4 (Remarks to the Author):

In this revision, the authors have done substantial amount of work to address the numerous reviewer's comments raised upon the previous submission. The manuscript has therefore been significantly improved. Though the majority of my prior comments have been addressed in the rebuttal, I have the following questions that require further clarifications.

1. It is not sufficiently clear what have been done in the capture-seq. The subtitle of "results" states "... lncRNAs are enriched at the synapse in the adult ILPFC" (line 98), which seems to suggest the profiling in a healthy control. However, in the following paragraph, it says the capture seq was done "on tissue derived from the adult mouse ILPFC of extinction trained mice" (lines 106-107). I am confused if the capture-seq was only done in fear extinction trained animals, or also carried out in control mice? Should a differential analysis of synapse capture-seq between control and extinction groups be a valuable approach to recognize candidate lncRNAs implicated in fear learning?

The synapse and nucleus capture-seq was performed using a pool of both retention control and extinction-trained mice. Our rationale for this was that capture-seq would serve as an entry point to investigate all lncRNAs that are enriched at the synapse regardless of condition. Once candidates were identified, we then performed RT-qPCR in a separate cohort of control vs extinction-trained animals to identify lncRNAs that are differentially expressed at the synapse following fear extinction learning. We acknowledge that differential analysis of synapse capture-seq between control and extinction groups may be a valuable approach to quickly recognize candidate lncRNAs implicated in fear extinction. However, simply relying on differential expression of lncRNAs at the synapse would not reveal the functional relevance of lncRNAs due to their multidimensional capacity to interact with proteins dynamically in a state dependent manner. The direct manipulation of lncRNA at the synapse is necessary to elucidate their role in the synaptic compartment.

We have edited the results (~line 106), and the manuscript now reads:

"..., we employed synaptosome isolation followed by lncRNA capture sequencing on tissue derived from the ILPFC of behaviourally trained adult mice (Figure 1a)...."

We have also edited the method section (~line ~784), which now reads:

"Mouse tissue and synaptosomes were collected from behaviourally trained adult mice and total RNA was isolated as described above...."

2. I appreciate the authors' effort to add RNAseq to the revision to complement the lncRNA profiling derived from the capture-seq in the initial approach. However, as the authors acknowledged (rebuttal letter page 17), none of the ten differential synaptic lncRNAs was significantly changed in the RNAseq analysis. This was highlighted as an advantage of using the targeted sequencing system. However, what was the sequencing depth of the RNAseq? I can not find this information in the manuscript, though I saw the read numbers of capture-seq were added to the revision. Would a deeper sequencing of RNAseq allow the authors to better compare the low-frequency events? Related, it does not appear the transcriptome RNAseq is added to the "methods" section. In the "lncRNA capture sequencing" part of "methods", it mentioned the NEBNext Ultra II RNA library prep kit for illumina was employed (line 785). Does that kit have two modules, one for polyA RNA isolation, and the other for rRNA depletion? If so, which one was applied in the study? This may affect the population of lncRNA that was collected.

We used the rRNA depletion kit and random primers to generate cDNAs for the lncRNA capture sequencing dataset. We have also added the synaptosome RNA-seq section (~line

800-808) and edited the capture-seq (~line 783-798) and sequencing data analysis (~line 811-829) in methods section, which now reads:

“lncRNA Capture Sequencing

Mouse tissue and synaptosomes were collected from behaviourally trained adult mice and total RNA was isolated as described above. 100 – 500 ng of rRNA depleted total synaptosome RNAs were used for library construction. The same amount of input was used for the nucleus-derived RNAs. cDNA libraries were generated with random primers using the NEBNext Ultra II RNA Library Prep Kit for Illumina (NEB) according to the manufacturer’s protocol. At least 1 µg of cDNA was used for subsequent capture. A custom-designed panel of 190,689 probes (Roche) targeting 28,228 known and predicted mouse lncRNAs¹, which was previously developed to improve the annotation of brain-enriched lncRNA, was used to capture amplified cDNA. The capture procedure was performed using the SeqCap EZ Hybridization and Wash Kit (Roche) and SeqCap EZ Accessory kit (Roche) according to the manufacturer’s instructions. Captured libraries were sequenced on an Illumina HiSeq 4000 platform with 150-bp paired-end reads (Genewiz). The sequencing depth for synaptosome samples was 56 million reads (8.4 Gbp) to 74 million reads (11.1 Gbp), with an average of 65.3 million reads, while the sequencing depth for the nucleus samples was 51 million reads (7.7 Gbp) to 128 million reads (19.2 Gbp), with an average of 82.4 million reads.

RNA Sequencing

Synaptosomes were collected, and total RNA was extracted as described above. 100 – 500 ng of total synaptosome RNA was used for library construction. cDNAs were generated using the SMARTer® Stranded Total RNA-Seq Kit v2 – Pico Input Mammalian (Takara). RNA-seq libraries were sequenced on an Illumina HiSeq 4000 platform with 150-bp paired-end reads (Genewiz). The sequencing depth for the synaptosome RC samples ranged from 37 million reads (5.5 Gbp) to 53 million reads (7.9 Gbp), averaging at 44.4 million reads. In contrast, the sequencing depth for the synaptosome EXT samples ranged from 47 million reads (7.1 Gbp) to 76 million reads (11.4 Gbp), with an average of 63.1 million reads.”

3. I have not found any descriptive information on the seven supplementary tables in the manuscript. Though I can guess, such as Table S4 appears to be the list of capture-seq probes, I am not certain about all (e.g., what is Table S5?). Some rows in a few tables (e.g., Table S1, S2, S3) are highlighted in color that should be explained as well. It says in the manuscript that the capture-seq system employed 190,689 probes, but there are only ~117K in table S4. Should these be consistent?

Table S4 lists the genomic coordinates for all capture regions from the lncRNA capture-seq experiment based on the mouse reference genome (mm10), while Table S5 shows only the genomic coordinates for Gas5 capture regions. In total, we used 190,689 probes to target 117,203 distinct capture regions. Certain captured regions may have multiple probes, and due to company proprietary, we do not have the exact probes coordinate used in this study.

We have added subheadings and descriptive title for each table, and the supplemental table titles are as follows:

Table S1: **Transcripts identified in the nucleus and synapse lncRNA capture-seq**

Table S2: **Synapse-enriched transcripts identified in the lncRNA capture-seq**

Table S3: **Transcripts identified in the synapse RC and synapse EXT RNA-seq**

Table S3: **Top10 capture-seq candidates identified in the synapse RC and EXT RNA-seq**

Table S4: **Genomic coordinates for all captured regions in the lncRNA capture-seq**

Table S5: **Genomic coordinates for the captured Gas5 region in the lncRNA capture-seq**

Table S6: **Gas5 RIP targets identified in the RC and EXT group**

Table S6: **Gas5 RIP targets identified exclusively in the RC and EXT group, and overlap in both**
Table S7: **Primers and CIRTS gRNA sequence.**

Reviewer #5 (Remarks to the Author):

The revised manuscript by Liao and colleagues has improved and become more clear, however several major weaknesses remain. Also, due to the rather minimalistic description of the experimental procedures, it is hard at times to evaluate the scientific validity or meaning of an experiment. The most important issues are described below:

1) Identification of the lncRNAs that are enriched in the synaptic compartment clearly constitute an important section of the manuscript. However, unfortunately the studies addressing this important question are poorly described, controlled and confusing.
a) Critical aspects of RNA sequencing-based studies are not covered in the manuscript. For example, how was the normalization done when comparing synaptic and nuclear compartments? Similarly, when RNAs from synaptic and nuclear compartments were compared by RT-qPCR, were the results normalized to 18S rRNA or PGK, as stated in the methods?

We used the tuxedo suite of protocols detailed in Pertea et al. 2016², and have also outlined our RNA sequencing-based analysis in the methods section (~line 810). We used StringTie to generate the FPKM value, which was normalized by the total number of sequenced fragments and the length of the transcript. To stabilize the variance, we used Ballgown's built-in functions to apply a log transformation and then fit standard linear models that can be used to test for differential expression.

We have edited the sequencing data analysis section (~line 811), which now reads:

“Sequencing Data Analysis

Cutadapt⁶² (v1.17, <https://cutadapt.readthedocs.io/en/stable/>) was used to trim low-quality nucleotides (Phred quality lower than 20) and Illumina adaptor sequences at the 3' end of each read for lncRNA capture sequencing data. Processed reads were aligned to the mouse reference genome (mm10) using HISAT2 (v2.1.0)⁶³. SAMtools (version 1.8)⁶⁴ was then used to convert “SAM” files to “BAM” files, remove duplicate reads, and sort and index the “BAM” files. To avoid the artefact signals potentially introduced by misalignments, we only kept properly paired-end aligned reads with a mapping quality of at least 20 for downstream analyses.

For capture-seq, three rounds of StringTie (v2.1.4)⁶⁵ were applied to i) perform reference-guided transcriptome assembly by supplying the GENCODE annotation file (V25) with the “-G” option for each sample, ii) generate a non-redundant set of transcripts using the StringTie merge mode, and iii) quantify the transcript-level expression for each sample, with the option of “-e -G merged.gtf”. Known protein-coding transcripts (with the GENCODE biotype as “protein_coding”) were removed from the StringTie results. Ballgown (v2.22.0)⁶⁶ was used to conduct transcript-level differential expression analysis. Alternative splicing analysis was performed using SUPPA2 (v2.3, <https://github.com/comprna/SUPPA>)⁷¹. Differential expression of Gas5 isoforms was created using a web-based visualization tool, IsoMix. Annotation reference and a file containing count data for each sample were used as input. IsoMix is available at <https://isomix.org/>.

For RNA-seq, the same pipeline prior to StringTie (v2.1.4)⁶⁵ analysis was conducted as described above. The gene annotation file (“merged.gtf”) used in capture-seq data analysis were supplied to StringTie (v2.1.4) to quantify the transcript-level expression for each sample,

with the option of “-e -G merged.gtf”, and generated the normalised abundance data (FPKM) for each transcript. Ballgown (v2.22.0)⁶⁶ was then used to conduct transcript-level differential expression analysis between the RC and EXT group.”

For RT-qPCR, we compared RNAs from synaptic and nuclear compartments by normalizing to 18S rRNA. The method has been edited (~line 677) and now reads:

“...Quantitative PCR was carried out on a RotorGeneQ (Qiagen) real-time PCR cyclor with SensiFAST SYBR master mix (Bioline) using primers for target genes. All transcript levels were normalized to 18S rRNA using the $\Delta\Delta CT$ method and each PCR reaction was run in duplicate for each sample and repeated at least twice.”

b) The lack of concordance of the standard bulk RNA-seq without enrichment for lncRNAs versus the results following capture-seq amplifies this concern. This lack of concordance could be due to inherent technical issues of enrichment protocols or normalization problems described above, which can affect both standard and enriched RNA-seq studies. This is particularly a concern since the studied lncRNAs (which didn't show concordant expression patterns) were those having the highest expression levels among the lncRNAs in the subset of interest and thus, were unlikely to be under-represented in standard bulk RNA-seq due to inefficient sampling (which can happen with transcripts with very low copy numbers).

We acknowledge that there is lack of concordance between the standard bulk RNA-seq and the capture-seq data, and that may be partly due to our enrichment protocols. We have therefore discussed the caveat of this approach and possible refinement in the discussion section (~line 403). The capture-seq experiment was performed using a pool of retention control and extinction-trained mice. As indicated above, we employed capture-seq as an entry point to investigate lncRNAs that are enriched at the synapse compared to the nucleus. In contrast, the bulk RNA-seq experiment was carried out separately using retention control and extinction-trained adult mice synaptosomes only. Hence, the bulk RNA-seq cannot be compared directly with the capture-seq result.

As the previous reviewer suggested, we used bulk RNA-seq to assess transcript-level changes between synapse RC and synapse EXT, and to corroborate the qPCR results depicted in Figure 1e-l. We chose 8 out of the top10 candidates for RT-qPCR verification and found that 6 out of the 8 candidates displayed a similar trend to the bulk RNA-seq result (i.e., higher expression in EXT compared to RC) although none of them passed the statistical threshold for differential expression at the transcriptome level. We also acknowledge that two lncRNAs (Rn7sk and Gm47305) showed discordance between the bulk RNA-seq and qPCR results. This lack of concordance may be due to potential misalignment of RNA-seq short read data, and because these two lncRNAs may not be fully annotated and similar to the Gas5 locus, additional transcripts may derive from these regions. We excluded small noncoding RNAs, Rny1 and Rny3, as they are less than 200 bp.

2) Another major section of the manuscript deals with defining the protein interactome of a GAS5 isoform that the authors believe is enriched in the synaptic compartment.

a) These studies are unfortunately performed using in vitro-transcribed RNAs incubated in cellular extracts. As RNAs fold and obtain their protein interactome co-transcriptionally, studies using in vitro-assembled systems are known to be highly prone to artifacts. The authors have attempted RIP studies to capture the interaction of their selected proteins with GAS5, however, there is no experimental details available about these studies. Whether the samples were crosslinked prior to lysis is not indicated and it seems that no RNA controls were used. In the absence of RNA pull down of endogenous GAS5, which has not been performed, RIP studies must be performed in a highly controlled manner to define the stoichiometry of the RNA-protein interactions described. Studies that merely show the presence of an interaction between two macromolecules can be misleading, since at some

level, many abundant RNAs interact with many abundant proteins at low stoichiometry (e.g. 5% of the copies of the relevant isoform of GAS5 may interact with a certain protein). These interactions can be captured by RIP-seq in the absence of appropriate controls, but they don't carry functional significance.

We acknowledge that our *in-vitro*-assembled system has caveats and may capture fortuitous interactions. Hence, a RIP assay using antibody against target protein is crucial to verify the interaction with Gas5. We observed an enrichment of Gas5 variant when we compare IgG control versus Caprin1 or G3bp2 pull down *in-vitro*. A similar enrichment was also observed for Caprin1 pull down *in-vivo*.

We have edited our RNA pulldown assay and included a paragraph of RIP in the methods section, which now reads:

“RNA Pull-Down Assay

Gas5 variant, deleted fragments, Neat1 and Adram were amplified using the T7 promoter sequence on the 5' end of the forward primers. The deleted Gas5, Neat1 and Adram DNA fragments were synthesized by IDT. The PCR products were gel extracted using the Gel DNA Recovery Kit (Zymo Research) and *in-vitro* transcribed using the HiScribe T7 Quick High Yield RNA Synthesis Kit (NEB) according to the manufacturer's instructions. The transcribed RNA was purified using the RNA Clean and Concentrator Kits (Zymo Research). Biotinylation and pull-down were performed using the Pierce Magnetic RNA-Protein Pull-Down Kit (Thermo Fisher) according to the manufacturer's instructions, and the concentration of biotinylated RNA was measured using a nanophotometer (IMPLEN). Briefly, ILPFC samples were incubated for 30 min on ice in NP40 cell lysis buffer (Thermo Fisher) supplemented with halt protease inhibitor (Thermo Fisher) and RNaseOUT RNase inhibitor (Thermo Fisher), and then centrifuged at 14,000 rpm for 10 min at 4°C. The supernatant was transferred to a new tube and protein concentration was measured using the Bradford assay (Sigma) on a nanophotometer (IMPLEN). 1 µg of biotinylated RNAs were incubated with Streptavidin beads for 30 min. The RNA-conjugated beads were washed three times and incubated with 500 µg of total ILPFC proteins for 1 h with rotation. RNA-protein-containing beads were UV-crosslinked, washed three times and subjected to either Western blot or mass spectrometry. For Western blot, the band intensity was measured using ImageJ.

RNA Immunoprecipitation

FLAG-tagged G3bp2 and Caprin1-expressing primary cortical neurons were crosslinked with 0.1% formaldehyde for 10 min, and incubated for 30 min on ice in NP40 cell lysis buffer (Thermo Fisher) supplemented with halt protease inhibitor (Thermo Fisher) and RNaseOUT RNase inhibitor (Thermo Fisher), and then centrifuged at 14,000 rpm for 10 min at 4°C. The supernatant was transferred to a new tube and protein concentration was measured using the Bradford assay (Sigma) on a nanophotometer (IMPLEN). 2 µg of IgG (cell signaling) or FLAG antibody (Sigma) was added to the pre-cleared lysate and incubated for 2 h with rotation. Protein G beads (Thermo Fisher) were added to each IP samples and incubate for another 1 h with rotation. Beads were washed three times, pelleted and resuspended in NucleoZOL (Macherey-Nagel). RNAs were purified and subjected to RT-qPCR as described in the previous section.

For the *in-vivo* RNA immunoprecipitation assay, ILPFC tissues from behavioural-trained animals were crosslinked with 0.1% formaldehyde for 10 min and homogenized in NP40 cell lysis buffer (Thermo Fisher) supplemented with halt protease inhibitor (Thermo Fisher) and RNaseOUT RNase inhibitor (Thermo Fisher), and incubated for 30 min on ice. Samples were then centrifuged at 14,000 rpm for 10 min at 4°C, and supernatant was transferred to a new tube and protein concentration was measured as described. 2 µg of Caprin1 antibody

(Proteintech) was added to the pre-cleared lysate and incubated and processed as described above.”

We also agree that the observed RNA-protein interaction may not carry functional significance and this is the reason why we performed Gas5 knockdown followed by live imaging of RNA granules to examine the effect of Gas5 on RNA granule trafficking and clustering.

b) As abundant RNA-binding proteins such as splicing factors and ribosomal factors bind any RNA to some extent, especially in an *in vitro*-assembled system such as the one used in the current study, ascribing functional significance to these likely fortuitous, concentration-driven interactions must be avoided (e.g. lines 193-208).

We agree that the functional significance of these interactions will need to be verified further in relevant biological contexts. That said, we are suggesting potential roles for Gas5 in regulating these biological processes and not ascribing functional significance to other interactions. We selected Caprin1 and G3bp2 for further analysis because of their known role in learning and memory, and because their interaction with Gas5 was further verified by reverse pulldown both *in-vitro* and under learning conditions.

c) The use of lncRNA ADRAM is certainly a step in the right direction. However, it's not sufficient and additional negative control lncRNAs (not snRNAs or tRNAs) should be used or even better, existing RNA pull down studies, many of which are available in GEO and SRA, can be queried for the pull down of Caprin1 and G3bp2 to determine their potential for forming nonspecific RNA interactions.

We have now added another negative lncRNA control, Neat1, and a scrambled RNA control to our existing RNA pull down studies. Both did not show binding to Caprin1 or G3bp2, and the manuscript now reads,

“ADRAM and Neat1, two nuclear lncRNAs involved in mediating epigenetic regulation¹⁰, exhibited no binding affinity for Caprin1 or G3bp2 (Supplemental Figure 5b and 5c).”

We have added the result in Supplemental Figure S5 and the primer sequences in Supplemental Table S7, and edited the legend, which now reads:

“Supplemental Figure 5. Blots displaying CAPRIN1 and G3BP2 proteins after incubating a) full-length *in-vitro* transcribed Gas5 or b) *Adram* or c) *Neat1* with ILPFC protein extracts. Input and scramble RNA control (5'-CCUGGUUUUUAAGGAGUGUCGCCAGAGUGCCG CGAAUGAAAAA-3') are indicated. d) Blots displaying CAPRIN1 and G3BP2 proteins after incubating different fragments of *in vitro* transcribed Gas5 with ILPFC protein extracts. Band intensity was quantified and plotted in Figure 4c and 4d.”

d) Lines 221-225: deletion mutagenesis studies must be validated with studies to prove that all deletion mutants are present in the extracts at the same level and that these mutations don't generate highly stable structures that may artifactually result in altered interactions with proteins.

We agree that the RNA structural state may alter the interactions with its target protein. However, since the study of RNA structural state is beyond the scope of this first observational study, we have suggested future experiments (~line 393), such as SHAPE-MaP and icSHAPE, to investigate how different proteins interact with the Gas5 modules. For the RNA pulldown assay, equal concentration of RNAs and proteins were added into our *in-vitro*-assembled system.

3) The functional studies using CIRTS is certainly innovative and exciting. However, there are a number of issues with these studies.

a) Unfortunately with only a single functional gRNA, off-target effects can't be ruled out, reducing the scientific rigor of the shown data.

We agree that, just like other RNA degrading system, there may be off-target effects. We chose one guide as it did not alter expression of other Gas5 variants in primary cortical neurons and the in-vitro cleavage assay (Supplemental Figure 6a and 6b).

We have edited the manuscript, which now reads (~line 264):

“..One of the guides degraded the *Gas5* variant ENSMUST00000162558.7 by more than 50% in both the nuclease cleavage assay and in primary cortical neurons, without affecting the expression of other *Gas5* variants and was therefore chosen for all subsequent knockdown experiments (Supplemental Figure 6).”

b) An important missing detail is whether the "scrambled" control RNA used in CIRTS studies is still a gRNA (with an intact RNA hairpin), or if it lacks the ability to associate with the hairpin binding protein in the CIRTS machinery.

The scrambled control is cloned into the same locus as the *Gas5* gRNA and therefore includes the intact RNA hairpin.

The method has been edited (~line 637) and now reads:

“...The U6-*Gas5* gRNA and U6-scrambled control gRNA was PCR amplified and inserted in the *Xba*I site...”

c) The studies in Fig. 5, while interesting, lack a positive control. Since the impact of changes in *GAS5* expression looks relatively modest, it would have been helpful to include a positive control (knock down of a known regulator of trafficking/assembly of G3bp2 containing condensates) to determine the biological significance of the observed changes.

This study is the first demonstration that a synapse enriched lncRNA variant can regulate the trafficking and clustering of RNA granules. The role of *Caprin1* and *G3bp2* in synaptic plasticity and memory formation has been demonstrated in Nakayama et. al. 2017³ and Kipper et. al. 2022⁴., and the regulator of assembly of G3bp2-containing condensates has been described in Kedersha et. al, 2016⁵. We are specifically looking at the effect of *Gas5* knockdown on G3bp2-containing granule and not the functional role of G3bp2 granule per se.

References

1. Bussotti G, et al. Improved definition of the mouse transcriptome via targeted RNA sequencing. *Genome Res* 26, 705-716 (2016).
2. Pertea M, Kim D, Pertea GM, Leek JT, Salzberg SL. Transcript-level expression analysis of RNA-seq experiments with HISAT, StringTie and Ballgown. *Nat Protoc* 11, 1650-1667 (2016).
3. Nakayama K, et al. RNG105/caprin1, an RNA granule protein for dendritic mRNA localization, is essential for long-term memory formation. *Elife* 6, (2017).
4. Kipper K, Mansour A, Pulk A. Neuronal RNA granules are ribosome complexes stalled at the pre-translocation state. *J Mol Biol* 434, 167801 (2022).
5. Kedersha N, et al. G3BP-Caprin1-USP10 complexes mediate stress granule condensation and associate with 40S subunits. *J Cell Biol* 212, 845-860 (2016)

REVIEWER COMMENTS

Reviewer #4 (Remarks to the Author):

To answer my prior question on the exact treatment condition of the mice used in the capture seq, the authors responded them to be “a pool of both retention control and extinction-trained mice”. However, in the manuscript, this was only updated as “behaviorally trained adult mice”. This is not sufficient as it does not provide necessary details. Based on table 1, it appears the capture seq had 6 nucleus replicates and 4 replicates in the synapse group. Does each of these replicates consist of a combination of both retention control and extinction-trained sample? If so, the composition of each behavioral condition in these samples should be provided. Furthermore, the authors have applied RNAseq, that was added in the first revision, to compare the Capture seq and RNA seq. However, the RNAseq was carried out in the synapse RNA samples only, with a separation of retention control and extinction-trained mice. I doubt it would be an effective validation as I thought it would be. I did not realize this until the authors provide additional details. It is pivotal for the authors to further clarify the manuscript and perhaps revised their conclusion remarks. I agree the capture seq has unique advantages, such as to detect low-abundance lncRNAs. However, it is intrinsically limited by its design. For example, based on the Ensembl database, there are currently many more Gas5 isoforms than the manuscript acknowledged. They are apparently not covered by capture seq in this study, though they could be examined by RNA seq.

I understand some regions are covered by multiple capture seq probes, which explained the discrepancy of probe number (190,689) and target number (117,203). But the authors should add this clarification to the manuscript.

Reviewer #5 (Remarks to the Author):

The revised manuscript authored by Liao and colleagues has improved in terms of clarity especially in the Methods section, however, unfortunately most major issues persist, as described below.

1) The normalization issues unfortunately don't allow a direct comparison between samples from different compartments. In all RNA-seq studies, the validity of normalization methods hinge on the assumption that the level of complexity of the transcriptome is largely similar between the samples being compared. This is true for both RPKM/FPKM or linear transformation of sequencing results and for more sophisticated approaches. While the information about the depth of sequencing of each compartment is not presented, it can be assumed that the nuclear compartment has a much more complex transcriptome than the synaptic one. If this is not the case, the authors need to provide clear data showing similar complexity. Otherwise, any meaningful comparison between the two compartments is not possible with the methods used by the authors. However, it can be asserted that certain RNAs were detectable in the synaptic compartment, which for the purposes of this manuscript is sufficient. Similarly, the validity of the PCR-based studies is also hinging on showing that the copy

number of “housekeeping” genes used in these studies is comparable in the different compartments and samples used in the study. Has this been ascertained? For example, can the level of housekeeping genes change in the synaptic compartment as a result of conditioning?

2) These shortcomings could have been addressed by in situ hybridization studies. Unfortunately in the data presented in Fig. 2G there is no clear evidence that the location of the red dots corresponds to synaptic compartment. There are red dots in many places in the images shown including in what appears to be the soma. Why a marker of synaptic densities, such as SV2A is not used?

3) In the RIP-seq studies unfortunately the key issues of significance and stoichiometry remain unaddressed, while the authors have added a much more clear experimental protocol which helps address several other issues.

4) In response to the point raised in the previous round of review “Lines 221-225: deletion mutagenesis studies must be validated with studies to prove that all deletion mutants are present in the extracts at the same level and that these mutations don't generate highly stable structures that may artifactually result in altered interactions with proteins”, the appropriate studies are checking the integrity of the mutant RNA (to show it's not degraded and that it exists in the mixture mostly in its full length form), and an in silico folding study to check for formation of super-stable structures. As the authors mentioned, a full scale RNA structural study is very much beyond the scope of the present manuscript, however, the simple controls listed above must be done for all mutational analyses (especially those adding or removing large chunks of sequence) to ensure that absence of signal is not stemming from degradation, multimerization or other consequences of formation of highly stable structures.

5) Unfortunately the potential for off-target effects with the CIRT studies remain, as only a single gRNA is used. This, at the least, must be clearly acknowledged in the Results and the Discussion. However, the uncertainty about the validity of the conclusions made with a single gRNA remains.

Minor points

6) Lines 128-138 discuss the presence of repeat elements, which don't seem to carry any significance beyond highly speculative statements by the authors. This section is best removed. lncRNAs are known to carry repeat element-derived sequences and barring an observed enrichment in the synaptic compartment, this section does not add anything to the study.

7) Line 153: “lncRNAs identified by capture-seq and subsequently validated by qPCR”. This statement is confusing. The capture-seq data was not validated by qPCR, as the qPCR study was done on distinct samples that did not correspond to those in capture-seq analysis.

8) Lines 163-165: this is because nascent transcripts are not spliced and is expected. It's best to remove this sentence which take away from the manuscript rather than add to it. Similarly, the statements about exon skipping (lines 160-163) and size of lncRNAs and number of exons (lines 166-169) either has

to be expanded or removed, as its significance and validity, as currently presented in the manuscript, is uncertain. Most of what is described is expected when cytoplasmic versus nuclear transcripts of all types (including protein-coding genes) are compared, and again, these discussions at the level presented take away from the manuscript rather than adding to it.

9) Line 173-176: is the Gas5 variant that is most highly enriched in the synaptic compartment according to figure 2e also the most abundant isoform in the cytoplasm? As written, it implies that the splicing pattern may have something to do with localization to synaptic compartment, whereas it might well be a simple matter of nuclear export efficiency. Without proving this, lines 181-184 are inaccurate. Also, the first part of the sentence starting at line 181 should be removed, as the presence of alternative splicing for both protein-coding and non-coding RNAs is well documented in the brain compartment and with the level of data shown in the manuscript, no conclusions beyond what is already known can be made. Also lines 184 -186 must be removed except if it can be proven that the variant of interest in this manuscript is not also the most abundant in the cytoplasm.

10) As discussed in the previous round of review “As abundant RNA-binding proteins such as splicing factors and ribosomal factors bind any RNA to some extent, especially in an in vitro-assembled system such as the one used in the current study, ascribing functional significance to these likely fortuitous, concentration-driven interactions must be avoided (e.g. lines 193-208)” The line numbers remain unchanged. Statements about the “GAS5 protein network” and any conclusions based on the unverified captured proteins and the data presented in Fig. 3 must be removed.

11) In a step in the right direction, the authors have added Neat1 to the in vitro-transcribed RNAs used in pull down. However, unfortunately the data do not appear in Figures 4C and D.

Reviewer #4 (Remarks to the Author):

To answer my prior question on the exact treatment condition of the mice used in the capture seq, the authors responded them to be “a pool of both retention control and extinction-trained mice”. However, in the manuscript, this was only updated as “behaviorally trained adult mice”. This is not sufficient as it does not provide necessary details.

We have edited the Results section (line ~106) and the manuscript now reads:

“...we employed lncRNA capture sequencing on replicate pools of nucleus and synaptosome fractions derived from the ILPFC of retention control (RC) and extinction (EXT) trained mice (Figure 1a).

Based on table 1, it appears the capture seq had 6 nucleus replicates and 4 replicates in the synapse group. Does each of these replicates consist of a combination of both retention control and extinction-trained sample? If so, the composition of each behavioral condition in these samples should be, provided.

We have added the composition of each behavioral condition in the Methods section (~line 773), which now reads:

“Nuclear and synaptosome fractions were collected from pooled ILPFC samples reflecting a total 24 RC and 16 EXT trained adult mice and total RNA was isolated as described above and 100-500 ng of rRNA depleted total synaptosome RNAs were used for library construction.”

Furthermore, the authors have applied RNAseq, that was added in the first revision, to compare the Capture seq and RNAseq. However, the RNAseq was carried out in the synapse RNA samples only, with a separation of retention control and extinction-trained mice. I doubt it would be an effective validation as I thought it would be. I did not realize this until the authors provide additional details. It is pivotal for the authors to further clarify the manuscript and perhaps revised their conclusion remarks. I agree the capture seq has unique advantages, such as to detect low-abundance lncRNAs. However, it is intrinsically limited by its design. For example, based on the Ensembl database, there are currently many more Gas5 isoforms than the manuscript acknowledged. They are apparently not covered by capture seq in this study, though they could be examined by RNA seq.

In response to the reviewer's previous comment, which noted a significant ratio of expression change in synapse-enriched lncRNAs during fear extinction learning (as indicated in Figure 1d), we included RNA-seq data in our initial revision to provide a more comprehensive examination of synaptic lncRNA expression during this process. The reviewer highlighted that 'five out of the eight top synapse-enriched lncRNAs are increased in fear extinction learning,' and suggested that 'a whole transcriptome RNA-seq may be able to address this'.

Accordingly, we have incorporated a whole transcriptome RNA-seq analysis to address this insightful suggestion and provide a more robust examination of the changes in synaptic lncRNA expression during fear extinction learning.

Our approach involved:

- i) identifying synapse-enriched lncRNAs using lncRNA capture sequencing data, comparing samples derived from synapse vs nucleus.
- ii) Employing RT-qPCR to specifically assess differential expression of the top synapse-enriched lncRNAs between synapse RC and synapse EXT (Figure 1e-i).

- iii) Conducting whole transcriptome RNA-seq experiment to examine the differential expression of gene/lncRNA between RC and EXT in the synaptic compartment, following reviewer's suggestion.

Although the whole transcriptome RNA-seq experiment does not directly validate the lncRNA capture sequencing experiment, it provides a valuable complementary dataset to the overall study of localized RNAs. We acknowledge the limitations of capture-seq in detecting certain lncRNAs due to incomplete annotation and coverage bias. We have now included these points, emphasizing the constraints and acknowledging the possibility we may have missed other lncRNAs at the synapse during fear extinction learning.

We have edited the discussion (~line 382), which now reads:

"The work presented here leverages the lncRNA capture-seq approach, specifically designed to enhance the identification of low-abundance lncRNAs that are often overlooked by lower resolution approaches like standard RNA-seq. Although capture-seq is efficient, it does however have inherent limitations. One notable caveat is its susceptibility to coverage bias due to a reliance on current, incomplete lncRNA annotations for probe targeting. This suggests that certain lncRNAs, including specific isoforms or entirely new entities, could go undetected. Recently released data based on the new version of the mouse genome (mm39), which identifies more isoforms, including additional Gas5 isoforms not mentioned in our study, highlights this limitation. Moreover, technical obstacles such as input requirement and the amount of RNA that is obtainable from the synaptic compartment constrain our ability to achieve cell-type resolution. Given the aforementioned incomplete lncRNA annotations, it is possible that other lncRNAs beyond those reported in this work might be present at the synapse during fear extinction learning."

We have also edited our result (~line 135 and ~line 149), which now reads:

"To determine if the expression of these synapse-enriched lncRNAs were altered by fear extinction training (EXT), we next selected 8 of the top 10 synapse-enriched lncRNAs for testing by RT-qPCR.

...Nonetheless, amongst the lncRNAs identified by capture-seq and subsequently shown to be upregulated by fear extinction training (EXT) by RT-qPCR, the stress-responsive lncRNA Gas5 attracted our attention as it has been implicated in the regulation of motivated behavior^{21,22}."

I understand some regions are covered by multiple capture seq probes, which explained the discrepancy of probe number (190,689) and target number (117,203). But the authors should add this clarification to the manuscript.

We have edited the Methods section (~line 778) for clarity, which now reads:

"A custom-designed panel of 190,689 probes (Roche) was used to target 28,228 known and predicted mouse lncRNAs, spanning 117,203 target regions. This panel, previously developed to improve the annotation of brain-enriched lncRNA, was used to capture amplified cDNA. Notably, some target regions are covered by multiple probes, explaining the discrepancy between the number of probes and the number of target regions."

Reviewer #5 (Remarks to the Author):

The revised manuscript authored by Liao and colleagues has improved in terms of clarity especially in the Methods section, however, unfortunately most major issues persist, as described below.

1) The normalization issues unfortunately don't allow a direct comparison between samples from different compartments. In all RNA-seq studies, the validity of normalization methods hinge on the assumption that the level of complexity of the transcriptome is largely similar between the samples being compared. This is true for both RPKM/FPKM or linear transformation of sequencing results and for more sophisticated approaches. While the information about the depth of sequencing of each compartment is not presented, it can be assumed that the nuclear compartment has a much more complex transcriptome than the synaptic one. If this is not the case, the authors need to provide clear data showing similar complexity. Otherwise, any meaningful comparison between the two compartments is not possible with the methods used by the authors. However, it can be asserted that certain RNAs were detectable in the synaptic compartment, which for the purposes of this manuscript is sufficient.

From our lncRNA capture-seq data, we detected a total of 28,164 transcripts originating from 19,157 gene loci in the nuclear compartment and 28,062 transcripts from 19,554 gene loci in the synaptic compartment. By applying a minimum expression threshold of FPKM > 0.5, as used in the EBI Expression Atlas (<https://www.ebi.ac.uk/gxa/FAQ.html>) and studies such as Wang et al. (Nature Communications, 2019, <https://www.nature.com/articles/s41467-019-12575-x>), we identified 22,974 transcripts (16,803 gene loci) in the nuclear compartment and 21,813 transcripts (15,089 gene loci) in the synaptic compartment. Both compartments exhibit similar levels of transcriptome complexity.

Furthermore, we categorized the FPKM into five categories: i) extremely low expression ($0.5 < \text{FPKM} \leq 1$), ii) low expression ($1 < \text{FPKM} \leq 5$), iii) moderate expression ($5 < \text{FPKM} \leq 50$), iv) high expression ($50 < \text{FPKM} \leq 100$), and v) extremely high expression ($\text{FPKM} > 100$). While we observed a comparable number of transcripts in each category, the synaptic compartment displayed a higher count of extremely low and extremely high expression transcripts. This suggests that numerous transcripts with extremely low expression in the synaptic compartment are challenging to capture using conventional RNA-seq and could potentially lead to an underestimation of transcriptional complexity within the synaptic compartment based on conventional RNA-seq studies. Please refer to Supplemental Figure 11a and the plot below for a visual representation of these five categories.

For transcripts with an FPKM > 0.5, we further examined the distribution of the number of isoforms per gene. Our findings indicate a comparable level of complexity at the gene locus level in terms of the number of identified isoforms. Please refer to Supplemental Figure 11b and the plot below for a visual representation of this distribution.

We have added the assessment of the complexity of the transcriptome in Supplemental Figure 11 and the Methods section (~line 817), which now reads:

“... We assessed the complexity of the transcriptome in both nuclear and synaptic compartments based on the total number of transcripts and the number of isoforms per gene. Because the level of transcriptome complexity is comparable between both compartments (Supplemental Figure 11), we therefore used Ballgown (v2.22.0) to conduct transcript-level differential expression analysis...”

We have also added the Supplemental Figure 11 legend, which now reads:

“Supplemental Figure 11. a) Distribution of FPKM counts from nucleus and synapse capture-seq data in five categories: i) extremely low expression ($0.5 < \text{FPKM} \leq 1$), ii) low expression ($1 < \text{FPKM} \leq 5$), iii) moderate expression ($5 < \text{FPKM} \leq 50$), iv) high expression ($50 < \text{FPKM} \leq 100$), and v) extremely high expression ($\text{FPKM} > 100$). b) Distribution of the number of isoforms per gene for transcripts with an $\text{FPKM} > 0.5$ in nucleus and synapse capture-seq data.”

Similarly, the validity of the PCR-based studies is also hinging on showing that the copy number of “housekeeping” genes used in these studies is comparable in the different compartments and samples used in the study. Has this been ascertained? For example, can the level of housekeeping genes change in the synaptic compartment as a result of conditioning?

We used the housekeeping gene, 18S rRNA, as it exhibited very stable expression across both the nucleus and synapse samples. The extinction training also did not alter the Ct values, and we have included the result in Supplemental Figure 2a.

We have also edited Supplemental Figure 2 legend, which now reads:

“...(a) Graph showing RT-qPCR Ct values of housekeeping gene 18S rRNA in the nucleus and synapse in the ILPFC following 60CS fear extinction training (EXT). Retention control (RC) is also indicated. $n = 4-6$ independent biological replicates per group. Statistical significance was determined using a two-tailed unpaired Student’s t-test...”

We have also edited Figure 1 legend, which now reads:

“...c) Classification of captured synaptic lncRNAs based on their genomic location with respect to protein coding genes and according to GENCODE V25 annotation. d) Bar plots showing the top 10 lncRNAs that are significantly enriched at the synapse and expressed as Fragments Per Kilobase of transcript per Million mapped reads (FPKM) ($n = 4-6$ independent biological replicates per group, two-tailed unpaired Student’s t-test. Rpph1, $t(8) = 3.608$, $p = 0.0069$; Rmrp, $t(8) = 3.146$, $p = 0.0137$; Rn7sk, $t(8) = 3.11$, $p = 0.0144$; Oip5os1, $t(8) = 3.907$, $p = 0.0045$; 9330121K16Rik, $t(8) = 4.443$, $p = 0.0022$; GM47305, $t(8) = 5.792$, $p = 0.0004$. (e-l) RT-qPCR of 8 of the 10 synapse-enriched candidates in the ILPFC following fear extinction training. 18S rRNA was used as the housekeeping gene for normalization (Supplemental Figure 2a)...”

2) These shortcomings could have been addressed by in situ hybridization studies. Unfortunately in the data presented in Fig. 2G there is no clear evidence that the location of the red dots corresponds to synaptic compartment. There are red dots in many places in the images shown including in what appears to be the soma. Why a marker of synaptic densities, such as SV2A is not used?

The signal in the soma is likely capturing Gas5 trafficking enroute to the synapse and there is very little Gas5 signal in the nucleus. Nonetheless, we have quantified the Gas5 puncta that co-localize with synaptic markers, PSD95 and SV2A, and now include these findings in Supplemental Figure 4.

We have edited the Results section (~line 173), which now reads:

“In addition, we also observed that the Gas5 variant co-localizes with PSD95 (-KCl, 79,3%; +KCl, 78.9%) and SV2A (-KCl, 62.7%; +KCl, 71.1%) in dendrites (Supplemental Figure 4). Taken together, the findings suggest that, in the adult brain, a specific Gas5 variant can localize to the synapse in an experience-dependent manner.”

We have added Supplemental Figure 4 legend, which now reads:

“Supplemental Figure 4. Representative image showing the co-localized expression of the Gas5 variant with the synaptic marker a) PSD95 and b) SV2A in primary cortical neurons. Representative images from $n \geq 8$ fields of view. Arrowheads show co-localized Gas5 expression at the dendritic spine. Scale bar, 20 μm . Red represents Gas5; magenta represents a) PSD95 or b) SV2A protein. The boxed region is enlarged in the inserts. Scale bar, 10 μm (PSD95), 5 μm (SV2A). ($n = 9$ -16 neurons per group, two-way ANOVA, $F_{1,58} = 259.3$ (a, bottom left), 4560 (a, bottom right), $p < 0.0001$; Dunnett's post hoc tests: nucleus -KCl versus dendrites -KCl, **** $p < 0.0001$, nucleus -KCl versus dendrites +KCl, **** $p < 0.0001$; $F_{1,36} = 137.4$ (b, bottom left), 403.5 (b, bottom right), $p < 0.0001$; Dunnett's post hoc tests: nucleus -KCl versus dendrites -KCl, **** $p < 0.0001$, nucleus -KCl versus dendrites +KCl, **** $p < 0.0001$. Error bars represent S.E.M.”

We have also edited the Methods section (~line 743), which now reads:

“...The co-localized Gas5 puncta in the nucleus and dendrites were quantified using Imaris (version 10.0.1).”

3) In the RIP-seq studies unfortunately the key issues of significance and stoichiometry remain unaddressed, while the authors have added a much more clear experimental protocol which helps address several other issues.

We have now added the negative control RNA, Neat1 and ADRAM, in Supplemental Figure 5a and 5b, showing that neither bind to Caprin1 or G3bp2.

We have also added a statement in the Result sections (~line 191), which now reads:

“In contrast, however, the nuclear lncRNAs ADRAM and Neat1 do not bind Caprin1 nor G3bp2 in primary cortical neurons.”

We have edited Supplemental Figure 5, which now reads:

“Gas5 RIP-qPCR of primary cortical neurons expressing FLAG-tagged a) Caprin1 or b) G3bp2.

Percentage of input is shown. Rabbit IgG control is used as control. Expression of lncRNA controls, ADRAM and Neat1, are indicated. n = 3 biological replicates”

4) In response to the point raised in the previous round of review “Lines 221-225: deletion mutagenesis studies must be validated with studies to prove that all deletion mutants are present in the extracts at the same level and that these mutations don’t generate highly stable structures that may artifactually result in altered interactions with proteins”, the appropriate studies are checking the integrity of the mutant RNA (to show it’s not degraded and that it exists in the mixture mostly in its full length form), and an *in silico* folding study to check for formation of super-stable structures. As the authors mentioned, a full scale RNA structural study is very much beyond the scope of the present manuscript, however, the simple controls listed above must be done for all mutational analyses (especially those adding or removing large chunks of sequence) to ensure that absence of signal is not stemming from degradation, multimerization or other consequences of formation of highly stable structures.

We have now performed an *in-vitro* folding assay and observe no formation of super-stable structures in all mutated RNAs.

In addition, after incubation with protein lysates for 2 hr, there is minimal RNA degradation for all mutated RNAs.

We have now added a statement in the Results section (~line 220), which reads:

“These mutations did not result in the formation of super-stable structures (Supplemental Figure 6a and b) and the mutant RNAs exhibited minimal degradation after incubation with protein lysates (Supplemental Figure 6c). ADRAM, a nuclear eRNA involved in mediating epigenetic regulation¹⁰, and Neat1, a nuclear lncRNA involved in paraspeckles formation²⁴, exhibited no binding affinity for Caprin1 or G3bp2 (Supplemental Figure 7b and c).”

We have also edited the Methods section (~line 842 and ~line 857), which now reads:

“... To test for the formation of highly stable structure, 10 µl of mutant RNAs were denatured at 95°C for 2 min, then transferred to ice for 1 min. 4 µl of ice-cold 5X RNA folding buffer (500 mM HEPES, pH 8.0; 500 mM NaCl) supplemented with RNaseOUT inhibitor (Invitrogen) was added and the RNA was then incubated for 15 min at 37°C to allow secondary structure formation. 2 µl of 100 mM MgCl₂ (pre-warmed at 37°C) was added and RNA was further incubated for 15 min at 37°C to allow tertiary structure formation. RNAs were visualized on a 1% native agarose gel

...The integrity of the mutant RNAs were also assessed after incubation with proteins as described above. 2 µl of Proteinase K (NEB) was added to the RNA-protein-beads mixture and incubated at room temperature for 10 min. The RNAs were purified using the RNA Clean and Concentrator Kits (Zymo Research) and visualized on a native agarose gel.”

We have added the Supplemental Figure 6 figure legend, which now reads:

“...RNA native gel (1%) displaying 1 µg of mutant (D1-D10) and full-length (FL) Gas5 RNAs a) before and b) after in-vitro RNA folding assay. Negative control RNAs, ADRAM and Neat1, are indicated. c) RNA native gel (1%) showing mutant (D1-D10) and full-length (FL) Gas5 RNAs isolated after incubating with ILPFC protein extracts for 2 hr. Negative control lncRNAs, ADRAM and Neat1, are indicated.”

5) Unfortunately, the potential for off-target effects with the CIRT studies remain, as only a single gRNA is used. This, at the least, must be clearly acknowledged in the Results and the Discussion. However, the uncertainty about the validity of the conclusions made with a single gRNA remains.

We have now added a statement in the Discussion section (line ~400), which reads:

“In addition, the Gas5 CIRT gRNA may exhibit off-target effects, which could be partially mitigated with the use of a second gRNA along with whole transcriptome sequencing to assess the integrity of other transcripts following targeted Gas5 knockdown.”

Minor points

6) Lines 128-138 discuss the presence of repeat elements, which don't seem to carry any significance beyond highly speculative statements by the authors. This section is best removed. lncRNAs are known to carry repeat element-derived sequences and barring an observed enrichment in the synaptic compartment, this section does not add anything to the study.

We agree with the reviewer and have substantially revised this section, although we retained a brief mention of the fact there was no enrichment for repeat element containing lncRNAs in synapse as we think there is value in reporting this observation.

The Results section (~line 129) now reads:

“Furthermore, while the majority of synapse-enriched lncRNAs (76.9%, 1987) contained putative transposable elements, including both short interspersed nuclear elements (SINEs) and long interspersed nuclear elements (LINEs) (Supplemental Table 2) Synapse-enriched lncRNAs did not exhibit enrichment of SINE or LINE elements compared to nucleus-enriched lncRNAs (Supplemental Table 2).”

7) Line 153: “lncRNAs identified by capture-seq and subsequently validated by qPCR”. This statement is confusing. The capture-seq data was not validated by qPCR, as the qPCR study was done on distinct samples that did not correspond to those in capture-seq analysis.

We have revised this sentence (~line 149) to improve clarity, which now reads:

“Nonetheless, amongst the lncRNAs identified by capture-seq and subsequently shown to be upregulated by fear extinction training (EXT) by RT-qPCR, the stress-responsive lncRNA *Gas5* attracted our attention as it has been implicated in the regulation of motivated behavior^{21,22}.”

8) Lines 163-165: this is because nascent transcripts are not spliced and is expected. It's best to remove this sentence which take away from the manuscript rather than add to it. Similarly, the statements about exon skipping (lines 160-163) and size of lncRNAs and number of exons (lines 166-169) either has to be expanded or removed, as its significance and validity, as currently presented in the manuscript, is uncertain. Most of what is described is expected when cytoplasmic versus nuclear transcripts of all types (including protein-coding genes) are compared, and again, these discussions at the level presented take away from the manuscript rather than adding to it.

As suggested by the reviewer, we have significantly revised and shortened this section. We have retained the observation of more retained introns in the nuclear RNA because this observation helps to further validate the validity of our nuclear and synaptic fractions.

9) Line 173-176: is the *Gas5* variant that is most highly enriched in the synaptic compartment according to figure 2e also the most abundant isoform in the cytoplasm? As written, it implies that the splicing pattern may have something to do with localization to synaptic compartment, whereas it might well be a simple matter of nuclear export efficiency. Without proving this, lines 181-184 are inaccurate. Also, the first part of the sentence starting at line 181 should be removed, as the presence of alternative splicing for both protein-coding and non-coding RNAs is well documented in the brain compartment and with the level of data shown in the manuscript, no conclusions beyond what is already known can be made. Also lines 184 -186 must be removed except if it can be proven that the variant of interest in this manuscript is not also the most abundant in the cytoplasm.

Nuclear export efficiency may be partially or largely driven by the splicing pattern. We think this is splitting hairs and that when an isoform is the most enriched in the synapse then it's a valid result. Nonetheless, we have edited these statements in the Results section (~line 175), which now reads:

“Taken together, the findings suggest that, in the adult brain, a specific *Gas5* variant can localize to the synapse in an experience-dependent manner.”

10) As discussed in the previous round of review “As abundant RNA-binding proteins such as splicing factors and ribosomal factors bind any RNA to some extent, especially in an in vitro assembled system such as the one used in the current study, ascribing functional significance to these likely fortuitous, concentration-driven interactions must be avoided (e.g. lines 193-208)” The line numbers remain unchanged. Statements about the “GAS5 protein network” and any conclusions based on the unverified captured proteins and the data presented in Fig. 3 must be removed.

We have removed these statements from the Results section.

11) In a step in the right direction, the authors have added Neat1 to the in vitro transcribed RNAs used in pull down. However, unfortunately the data do not appear in Figures 4C and D.

We have added Neat1 to Figure 4c and 4d.

We have modified the Figure 4 legend, which now reads:

“...Band intensity values of c) CAPRIN1 and d) G3BP2 are normalized to their undeleted full-length control and lncRNAs, ADRAM and Neat1, were used as negative controls, one-way ANOVA for Caprin1 ($F_{(11,24)} = 5.99$, $p = 0.0001$; Dunnett’s post hoc tests: D3 versus D9, $p = 0.0163$; D3 versus D10, $p = 0.0123$; D3 versus ADRAM, $p = 0.009$; D3 versus Neat1, $p = 0.0049$) and G3bp2 ($F_{(11,24)} = 6.438$, $p = 0.0006$; Dunnett’s post hoc tests: D6 versus D5, $p = 0.0128$; D6 versus D10, $p = 0.0017$; D6 versus ADRAM, $p = 0.001$; D6 versus Neat1, $p = 0.0007$). * $p < 0.05$, ** $p < 0.01$, *** $p < 0.005$. Error bars represent S.E.M.”

We have also edited the result section (~line 200), which now reads:

“ADRAM, a nuclear eRNA involved in mediating epigenetic regulation¹⁰, and Neat1, a nuclear lncRNA involved in paraspeckles formation²⁴, exhibited no binding affinity for Caprin1 or G3bp2 (Supplemental Figure 7b and c)”.

REVIEWERS' COMMENTS

Reviewer #5 (Remarks to the Author):

The revised manuscript has significantly improved and while shortcomings remain, the added data allows the reader to judge the merits of the conclusions in the manuscript. However, the analyses related to the issue raised in point 2 about quantitation of the Gas5 puncta needs to be redone. Including SV2A and PSD95 staining is certainly very helpful, however, what needs to be quantitated is the ratio of Gas5 puncta colocalized with these two marker proteins vs all the rest of the non-colocalized Gas5 puncta in the entire cell. The number of cells/animals used in this quantitation study should be indicated.

REVIEWERS' COMMENTS

Reviewer #5 (Remarks to the Author):

The revised manuscript has significantly improved and while shortcomings remain, the added data allows the reader to judge the merits of the conclusions in the manuscript. However, the analyses related to the issue raised in point 2 about quantitation of the Gas5 puncta needs to be redone. Including SV2A and PSD95 staining is certainly very helpful, however, what needs to be quantitated is the ratio of Gas5 puncta colocalized with these two marker proteins vs all the rest of the non-colocalized Gas5 puncta in the entire cell. The number of cells/animals used in this quantitation study should be indicated.

We have repeated the analysis of Supplementary Figure 4 with quantitation of Gas5 puncta colocalized with PSD95 or SV2A as a ratio vs total non-colocalized Gas5 puncta.

We have also included the number of neurons used in this quantification study in Supplementary Figure 4 legend.

We have edited the results (~line 174) and the manuscript now reads:

“...In addition, we also observed that the Gas5 variant co-localizes with PSD95 (ratio of co-localized dendritic puncta: -KCl, 5.76; +KCl, 3.02) and the dendritic marker, SV2A (ratio of co-localized dendritic puncta: -KCl, 2.46, +KCl, 2.98) in dendrites (Supplemental Figure 4)...”

We have edited the Supplementary Figure 4 legend, which now reads:

“Representative image showing the co-localized expression of the Gas5 variant with the synaptic marker a) PSD95 and b) SV2A in primary cortical neurons. Arrowheads show co-localized Gas5 expression at the dendritic spine. Scale bar, 20 μm . Red represents Gas5; magenta represents a) PSD95 or b) SV2A protein. The boxed region is enlarged in the inserts. Scale bar, 10 μm (PSD95), 5 μm (SV2A). Graph showing ratio of Gas5 puncta colocalized with a) PSD95 (-KCl, n = 15 neurons, +KCl, n = 16 neurons, two-way ANOVA, $F_{1,58} = 182.1$, $p < 0.0001$; Dunnett's post hoc tests: nucleus -KCl versus dendrites -KCl, **** $p < 0.0001$, nucleus -KCl versus dendrites +KCl, **** $p < 0.0001$) or b) SV2A (-KCl, n = 9 neurons, +KCl, n

= 11 neurons, two-way ANOVA, $F_{1,36} = 59.31$, $p < 0.0001$; Dunnett's post hoc tests: nucleus - KCl versus dendrites -KCl, $***p < 0.001$, nucleus -KCl versus dendrites +KCl, $****p < 0.0001$) versus total number of non-localized puncta. Error bars represent S.E.M."